# Theoretical Constraints on the Expressive Power of RoPE-based Tensor Attention Transformers

## Abstract

Tensor Attention extends traditional attention mechanisms by capturing high-order correlations across multiple modalities, addressing the limitations of classical matrix-based attention. Meanwhile, Rotary Position Embedding (RoPE) has shown superior performance in encoding positional information in long-context scenarios, significantly enhancing transformer models' expressiveness. Despite these empirical successes, the theoretical limitations of these technologies remain underexplored. In this study, we analyze the circuit complexity of Tensor Attention Transformer and extend to its RoPE-based Tensor Attention variants, showing that with polynomial precision, constant-depth layers, and linear or sublinear hidden dimension, they cannot solve fixed membership problems or $(A_{F,r})^*$ closure problems, under the assumption that $\mathsf{TC}^0 \neq \mathsf{NC}^1$. These findings highlight a gap between the empirical performance and theoretical constraints of Tensor Attention and RoPE-based Tensor Attention Transformers, offering insights that could guide the development of more theoretically grounded approaches to Transformer model design and scaling.

## 1 Introduction

Large Language Models (LLMs), such as OpenAI's ChatGPT (Achiam et al., 2023), Google's Gemini (Google, 2024), Anthropic's Claude 3.5 (Anthropic, 2024), and Meta's LLaMA 3.3 (LT, 2024) have reshaped a wide range of fields by demonstrating unprecedented advancements. These advancements are primarily due to their capability to efficiently process long-context inputs, a crucial feature for tasks like summarizing lengthy documents (e.g., medical reports, legal analyses, technical briefs), enabling superior reasoning and problem-solving performance at a level comparable to expert human analysis. At the core of these advancements lies the Transformer architecture (Vaswani et al., 2017), driven by its self-attention mechanism. Understanding computational primitives that Transformer components enable is pivotal for principled interpretations and exposing limitations in Transformer-based systems.

Previous research has investigated these questions by analyzing the expressiveness of Transformers. As an illustration, the work in (Merrill & Sabharwal, 2023) showed that constant-depth threshold circuit families can effectively emulate Transformers with precision $c \log n$ and depth-$d$. This holds true in both non-uniform and L-uniform computational models. This result highlights Transformers' computational efficiency and structural adaptability when analyzed through circuit complexity theory's lens. Expanding on these results, (Chiang, 2024) showed that Transformers with $O(\log n)$ precision belong to DLOGTIME-uniform $\mathsf{TC}^0$, even when the absolute error is bounded by $2^{-O(\mathrm{poly}(n))}$.

To augment the capabilities of Transformers, innovations such as Rotary Position Embedding (RoPE)(Su et al., 2024) have been proposed. Through the rotation matrices, RoPE improves the sequence length adaptability while enhancing the efficacy of attention mechanisms. Meanwhile, multi-view approaches are increasingly recognized for capturing high-order correlations in diverse data types, including mathematical data (Sanford et al., 2024), graph structures (Demirel et al., 2021; Luo et al., 2023), and multi-modality datasets (Lahat et al., 2015). Models like GPT-4o (OpenAI, 2024) and Google's Project Astra (Google, 2024) exemplify this trend, integrating reasoning across

multi-modality in real-time. Despite these advancements, classical attention mechanisms face representational limitations. Specifically, (Sanford et al., 2024) demonstrated that matrix attention can only capture pairwise correlations, falling short in modeling triple-wise or higher-order interactions. Addressing such limitations typically requires multiple layers or carefully designed architectures, complicating the integration of multi-view information.

To overcome these constraints, (Sanford et al., 2024) and (Alman & Song, 2024) proposed Tensor Attention, a higher-order extension of matrix attention. Tensor Attention intrinsically captures high-order correlations, defined as $\mathsf{Softmax}(Q(K_1 \oslash K_2)^\top)(V_1 \oslash V_2)$ (see Definition 2.20), where $\oslash$ denotes the column-wise Kronecker product (see Definition 2.13). Here, $Q$, $K_1/V_1$, and $K_2/V_2$ represent inputs from different views or modalities. This raises a natural question: *Does the* RoPE *and tensor attention enhance the expressiveness of the* RoPE-*based tensor attention Transformer?*

This work addresses this question through the lens of circuit complexity, advancing the theoretical understanding of tensor attention and RoPE-based tensor attention mechanisms. We present a rigorous analysis of tensor attention Transformers and RoPE-based tensor attention Transformers, delineating their intrinsic computational limitations. Our approach methodically evaluates the circuit complexity of each architectural component, ranging from basic trigonometric operations to the comprehensive RoPE-based tensor attention Transformers. Specifically, it is demonstrated that uniform $\mathsf{TC}^0$ circuits are amenable to simulating the components mentioned above. Furthermore, it is proven that, unless $\mathsf{TC}^0 = \mathsf{NC}^1$, tensor attention Transformers, as well as RoPE-enhanced tensor attention Transformers with $O(1)$ layers, $\mathrm{poly}(n)$-precision, and a feature dimension $d = O(n)$ are incapable of solving fixed membership problems or $(A_{F,r})^*$ closure problems. This finding underscores fundamental expressivity constraints inherent to tensor attention and RoPE-based tensor attention architectures.

The summary of our contributions to the theoretical understanding of these architectures and their computational boundaries, rooted in circuit complexity theory, showed as follows:

- We demonstrate that a DLOGTIME-uniform $\mathsf{TC}^0$ circuit family can simulate a tensor attention Transformer with constant depth, $\mathrm{poly}(n)$ size, and $\mathrm{poly}(n)$ precision. Then we extend the result to RoPE-based tensor attention Transformer. (Based on Theorem 3.7 and Theorem 4.5).

- We demonstrate that, unless $\mathsf{TC}^0 = \mathsf{NC}^1$, a tensor attention Transformer or a RoPE-based tensor attention Transformer with $O(1)$ layers, $\mathrm{poly}(n)$ precision, and a feature dimension $d = O(n)$ are incapable of accomplishing the fixed membership problems (Based on Theorem 5.9).

- We demonstrate that, unless $\mathsf{TC}^0 = \mathsf{NC}^1$, a tensor attention Transformer or a RoPE-based tensor attention Transformer with $O(1)$ layers, $\mathrm{poly}(n)$ precision, and a feature dimension $d = O(n)$ are incapable of accomplishing the $(A_{F,r})^*$ closure problems (Based on Theorem 5.10).

## 2 PRELIMINARY

This section establishes the essential concepts and definitions. Section 2.1 provides an in-depth exploration of float point number computation. Section 2.2 offers a comprehensive overview of computational complexity classes. Then, Section 2.3 presents essential techniques employed in tensor operations. Finally, Section 2.4 explores the fundamental components that constitute the RoPE-based tensor attention Transformers.

**Notations.** Let $n \in \mathbb{Z}_+$ represent any positive integer. The set of the first $n$ natural numbers is denoted as $[n] := \{1, 2, \ldots, n\}$. The inner product of vectors $\alpha, \beta \in \mathbb{R}^n$ is given by $\langle \alpha, \beta \rangle$. The vector $\mathbf{1}_n$ is an $n$-dimensional vector, where each component is one. The $\ell_\infty$ norm of a matrix $W \in \mathbb{R}^{n \times d}$ is represented as $\|W\|_\infty := \max_{m \in [n], n \in [d]} |W_{m,n}|$. Finally, a binary string $x_i \in \{0, 1\}^*$ denotes a sequence of arbitrary length.

### 2.1 FLOAT POINT OPERATIONS

We present basic concepts of the computational foundation.

**Definition 2.1** (Float point number, Definition 9 from (Chiang, 2024)). *A $p$-bit floating-point number is a pair $\langle m, e \rangle$, where $m$ (the significand) and $e$ (the exponent) are integers satisfying $|m| \in \{0\} \cup [2^{p-1}, 2^p)$ and $e \in [-2^p, 2^p)$. The real value represented by $\langle m, e \rangle$ is $m \cdot 2^e$. We denote by $\mathbb{F}_p$ the set of all such $p$-bit floating-point numbers.*

Then, we introduce the rounding operation, which is necessary for floating point number computation in modern computers.

**Definition 2.2** (Rounding, Definition 9 from (Chiang, 2024)). *Given any real number or float point value $x$, the notation $\mathrm{round}_p(x)$ denotes the $p$-bit float point number closest to $x$. In cases where we have different numbers equidistant from $x$, the tie-breaking convention dictates that $\mathrm{round}_p(x)$ will be the even significand one.*

Floating-point operations can also be computed efficiently with logical circuits. Details of these theoretical results are deferred to Appendix B.

## 2.2 CIRCUIT COMPLEXITY

In computational theory, a Boolean circuit, constructed using basic gates such as AND, OR, and NOT, represents a core model of computation. A precise mathematical definition of this structure comes below.

**Definition 2.3** (Boolean Circuit, Definition 6.1 in (Arora & Barak, 2009)). *An $n$ variables Boolean circuit is defined as $C_n : \{0, 1\}^n \rightarrow \{0, 1\}$ and is depicted by a directed acyclic graph (DAG). In this representation, logical gates such as AND, OR, and NOT correspond to the vertices of the graph. Input vertices, each linked to one of the $n$ Boolean variables, have an in-degree of 0, whereas non-input vertices derive their values from outputs of preceding gates in the structure.*

Based on the boolean circuit, we can define the recognizable languages.

**Definition 2.4** (Languages, Definition 6.2 from (Arora & Barak, 2009)). *A Boolean circuit family $\mathcal{C}$ is said to recognize language $L \subseteq \{0, 1\}^*$ if a Boolean circuit $C_{|z|} \in \mathcal{C}$ with $|z|$ variables exists, s.t., $C_{|z|}(z) = 1$, iff $z \in L$, for every string $z \in \{0, 1\}^*$.*

Then, we can define different circuit complexity classes based on the language we defined above.

**Definition 2.5** ($\mathsf{NC}^i$, Definition 6.21 from (Arora & Barak, 2009)). *The class $\mathsf{NC}^i$ is defined as the set of languages that are recognizable using Boolean circuits of size $O(\mathrm{poly}(n))$ and depth $O((\log n)^i)$, with logical gates of bounded fan-in, including NOT, OR, and AND gates.*

When Boolean circuits are permitted to incorporate gates such as AND and OR with unbounded fan-in, their ability to process languages becomes significantly enhanced. The development leads to the complexity classes of $\mathsf{AC}^i$.

**Definition 2.6** ($\mathsf{AC}^i$, Definition 6.22 from (Arora & Barak, 2009)). *Languages which can be computed by the Boolean circuit of depth $O((\log n)^i)$, size $O(\mathrm{poly}(n))$, unbounded fan-in gates, including AND, OR, NOT, are contained in the class $\mathsf{AC}^i$.*

The MAJORITY gates can simulate AND, NOT, OR gates, which yield an output of 1 if the majority of inputs are 1, and 0 otherwise. By incorporating MAJORITY gates, one can define a broader complexity class known as $\mathsf{TC}^i$.

**Definition 2.7** ($\mathsf{TC}^i$, Definition 4.34 from (Vollmer, 1999)). *If we have languages are recognizable by $O(\mathrm{poly}(n))$ size Boolean circuits of $O((\log n)^i)$ depth, and unbounded fan-in gates, including MAJORITY, NOT, OR, and AND gates. If half of the inputs are 1, the MAJORITY gate will output 1.*

*The class $\mathsf{TC}^i$ contains languages that are recognizable by Boolean circuits of size $O(\mathrm{poly}(n))$, depth $O((\log n)^i)$, and gates with unbounded fan-in, including NOT, OR, AND, and MAJORITY gates. A MAJORITY gate outputs one if more than half of its inputs are one.*

As Definition 2.7 shows, MOD or THRESHOLD gates (for prime moduli) can replace MAJORITY gates. Boolean circuits employing such gates are collectively referred to as threshold circuits. Next, we formally introduce the class P.

**Definition 2.8** (P, Definition 1.20 from (Arora & Barak, 2009)). *A language is considered to be in* P *if it can be decided by a deterministic Turing machine within polynomial time of input size.*

The hierarchical relationships among certain circuit families are encapsulated in the following well-known result.

**Fact 2.9** (Corollary, Corollary 4.35 from (Vollmer, 1999)). *Any $i \in \mathbb{N}$, the following inclusions are valid:* $\mathsf{NC}^i \subseteq \mathsf{AC}^i \subseteq \mathsf{TC}^i \subseteq \mathsf{NC}^{i+1} \subseteq \mathsf{P}$.

If $i = 0$, it has been established $\mathsf{NC}^0 \subsetneq \mathsf{AC}^0 \subsetneq \mathsf{TC}^0$. However, it remains unresolved whether $\mathsf{TC}^0 \subsetneq \mathsf{NC}^1$. Moreover, the question of whether $\mathsf{NC} := \cup_{i \in \mathbb{N}} \mathsf{NC}^i \subsetneq \mathsf{P}$ is an open problem. Additional details can be found in Corollary 4.35 from (Vollmer, 1999). Non-uniform circuit families, characterized by their lack of consistent structural design across varying input sizes, are theoretically capable of addressing undecidable problems. Nevertheless, their impracticality arises from the infinite length required for their description. In contrast, Uniform circuit families, which adhere to a systematic computational model, hold greater relevance in the study of complexity and formal language theory. We begin with the definition of L-uniformity.

**Definition 2.10** (L-uniformity class, Definition 6.5 from (Arora & Barak, 2009)). *Denote* C *as a class of languages represented by circuit family $\mathcal{C}$ (such as, $\mathsf{NC}^i$, $\mathsf{AC}^i$, or $\mathsf{TC}^i$). A language $L \subseteq \{0,1\}^*$ is classified as belonging to the* L*-uniform class of* C *if existing a Turing machine can map $1^n$ to $\mathcal{C}$ class circuit with $n$ variables in $O(\log n)$ space, for each $n \in \mathbb{N}$, and the resulting circuit $C_n$ recognizes $L$.*

Then, the DLOGTIME-uniformity and examine its correspond to L-uniformity will be introduced.

**Definition 2.11** (DLOGTIME-uniformity, Definition 4.28 from (Barrington & Immerman, 1994)). *Let* C *be a class of languages represented by circuit family $\mathcal{C}$ (such as $\mathsf{NC}^i$, $\mathsf{AC}^i$, or $\mathsf{TC}^i$). A language $L \subseteq \{0,1\}^*$ is defined to belong to the* DLOGTIME*-uniform class of* $\mathcal{C}$ *if a random-access Turing machine can map $1^n$ to $n$ variables circuit $C_n$ in $\mathcal{C}$ within $O(\log n)$ time, for every $n \in \mathbb{N}$, such that $C_n$ recognizes $L$.*

The concept of DLOGTIME-uniformity aligns with that of L-uniformity, except in smaller circuit classes that do not have the capability to imitate the constructing machine. Further exploration of uniformity concepts can be found in (Barrington & Immerman, 1994; Hesse et al., 2002). Within this paper, references to uniform $\mathsf{TC}^0$ pertain specifically to DLOGTIME-uniform $\mathsf{TC}^0$.

## 2.3 Tensor Operation Analysis Techniques

We first define operations such as the Kronecker product, a matrix operation that takes two matrices of any size and produces a block matrix. Unlike standard matrix multiplication, it is useful for introducing and analyzing tensor attention. Then, we introduce some key techniques for applying tensor attention to RoPE.

**Definition 2.12** ($\otimes$ Kronecker product). *Given $K_1 \in \mathbb{R}^{n_1 \times d_1}$ and $K_2 \in \mathbb{R}^{n_2 \times d_2}$, let $K := K_1 \otimes K_2 \in \mathbb{R}^{n_1 n_2 \times d_1 d_2}$ be defined for any $i_1 \in [n_1], j_1 \in [d_1]$ and $i_2 \in [n_2], j_2 \in [d_2]$ as $K_{i_1 + (i_2-1)n_1, j_1 + (j_2-1)d_1} = (K_1)_{i_1, j_1} \cdot (K_2)_{i_2, j_2}$.*

**Definition 2.13** ($\oslash$ column-wise Kronecker product). *Given matrices $K_1 \in \mathbb{R}^{n_1 \times d}, K_2 \in \mathbb{R}^{n_2 \times d}$, we define matrix $K := K_1 \oslash K_2 \in \mathbb{R}^{n_1 n_2 \times d}$ as for any $i_1 \in [n_1], i_2 \in [n_2], j \in [d]$, $K_{i_1 + (i_2-1)n_1, j} := (K_1)_{i_1, j} \cdot (K_2)_{i_2, j}$.*

**Definition 2.14** ($\ominus$ row-wise Kronecker product). *Given matrices $K_1 \in \mathbb{R}^{n \times d_1}, K_2 \in \mathbb{R}^{n \times d_2}$, we define matrix $K := K_1 \ominus K_2 \in \mathbb{R}^{n \times d_1 d_2}$ as for any $\forall i \in [n], j_1 \in [d_1], j_2 \in [d_2]$, $K_{i, j_1 + (j_2-1)d_1} := (K_1)_{i, j_1} \cdot (K_2)_{i, j_2}$.*

## 2.4 Transformer Block

With the mathematical foundation in place, this section outlines the key components of the RoPE-based tensor attention Transformers architecture, starting with the rotation matrix. This fundamental building block is generalized to encode the relative positions within a sequence, facilitating the embedding of positional context.

**Definition 2.15** (Rotation matrix). *Noted $j$ represents position index within input sequence and $i$ denotes token index. The relative rotation matrix is then expressed as:*

$$R_{j-i} = \begin{bmatrix} R((j-i)\theta_1) & 0 & \cdots & 0 \\ 0 & R((j-i)\theta_2) & \cdots & 0 \\ \vdots & \vdots & \ddots & \vdots \\ 0 & 0 & \cdots & R((j-i)\theta_{d/2}) \end{bmatrix},$$

*where the angular frequencies $\theta_1, \cdots, \theta_{d/2}$ are all predefined. More about selecting $\theta$, consult Equation (15) from (Su et al., 2024).*

Leveraging rotation matrices mentioned above, RoPE-based tensor attention embeds positional relation intrinsically within the computational process of attention. Now, we are about to introduce the RoPE-based tensor attention. First, we introduce the parameters and input.

**Definition 2.16** (Input and weight matrix). *We define the input sequence as $X \in \mathbb{R}^{n \times d}$ and the key, query, and value weight matrix as $W_{K_1}, W_{K_2}, W_Q, W_{V_1}, W_{V_2} \in \mathbb{R}^{d \times d}$. Then, we define the key, query, and value matrix as $K_1 := XW_{K_1} \in \mathbb{R}^{n \times d}$, $K_2 := XW_{K_2} \in \mathbb{R}^{n \times d}$, $Q := XW_Q \in \mathbb{R}^{n \times d}$, $V_1 := XW_{V_1} \in \mathbb{R}^{n \times d}, V_2 := XW_{V_2} \in \mathbb{R}^{n \times d}$.*

Then, based on Definition 2.13, we define RoPE-based tensor attention matrix in the following way.

**Definition 2.17** (RoPE-based tensor attention). *As we defined in Definition 2.15 and 2.16. We compute the new attention matrix $A \in \mathbb{F}_p^{n \times n^2}$ by:*

$$A_{j_1, j_2+(j_3-1)d} := (\exp(Q_{j_1,*} R_{j_1, j_2+(j_3-1)d} \cdot (K_{*, j_2+(j_3-1)d})^\top / d))_{j_1, j_2+(j_3-1)d},$$

*where*

$$R_{j_1, j_2+(j_3-1)d} = R_{j_1-j_2} \ominus R_{j_1-j_3} \in \mathbb{F}_p^{n \times n} \quad \text{and} \quad K = K_1 \otimes K_2 \in \mathbb{F}_p^{n^2 \times d^2}.$$

**Definition 2.18** (Single RoPE-based tensor attention layer, Definition 7 in (Sanford et al., 2024), Definition 1.1 in (Alman & Song, 2024), Definition 3.8 in (Liang et al., 2024)). *Given input matrices $Q, K_1, K_2, V_1, V_2 \in \mathbb{F}_p^{n \times d}$, $R \in \mathbb{F}_p^{d \times d}$ as Definition 2.17, and attention matrix $A \in \mathbb{F}_p^{n \times n^2}$ as Definition 2.17, we compute the $i$-th RoPE-based tensor attention layer $\mathsf{Attn}_i$ as*

$$\mathsf{Attn}_i(X) := \underbrace{D^{-1}}_{n \times n} \underbrace{A}_{n \times n^2} \underbrace{V}_{n^2 \times d}$$

*where $D := \mathrm{diag}(A\mathbf{1}_{n^2}) \in \mathbb{F}_p^{n \times n}$, and $V = V_1 \oslash V_2 \in \mathbb{F}_p^{n^2 \times d}$.*

For derivation convenience, we also introduce an equivalent form of a single RoPE-based tensor attention layer, which follows directly from Fact D.1.

**Fact 2.19** (Equivalent form of single RoPE-based tensor attention layer). *The single RoPE-based tensor attention layer in Definition 2.18 can be equivalently written as*

$$\mathsf{Attn}_i(X) := D^{-1} A(X \otimes X)(W_{V_1} \oslash W_{V_2}).$$

Then, we introduce a single tensor attention layer.

**Definition 2.20** (Single tensor attention layer, Definition 7 in (Sanford et al., 2024), Definition 1.1 in (Alman & Song, 2024), Definition 3.5 in (Liang et al., 2024)). *Given input matrices $Q, K_1, K_2, V_1, V_2 \in \mathbb{F}_p^{n \times d}$, compute the following matrix $\mathsf{Attn}_i(X) := D^{-1}AV$, where (1) $A := \exp(QK^\top / d) \in \mathbb{F}_p^{n \times n^2}$ and $K := K_1 \oslash K_2 \in \mathbb{F}_p^{n^2 \times d}$, (2) $D := \mathrm{diag}(A\mathbf{1}_{n^2}) \in \mathbb{F}_p^{n \times n}$, and (3) $V := V_1 \oslash V_2 \in \mathbb{F}_p^{n^2 \times d}$.*

Next, we can also integrate multi-layer attention and the additional mechanism mentioned above to construct a comprehensive Transformer.

**Definition 2.21** (Multiple layer tensor attention Transformer). *The number of Transformer's layers is denoted by $m$. In the $i$-th Transformer layer, let $g_i$ signify components distinct from self-attention, where $g_i : \mathbb{F}_p^{n \times d} \to \mathbb{F}_p^{n \times d}$, each $i \in [m]$. And $\mathsf{Attn}_i$ represent $i$-th layer attention mechanism(as*

*defined in Definition 2.18 and Definition 2.20). Given an input data matrix $X \in \mathbb{F}_p^{n \times d}$, an $m$-layer Transformer* $\mathsf{TF} : \mathbb{F}_p^{n \times d} \to \mathbb{F}_p^{n \times d}$ *is formally defined as:*

$$\mathsf{TF}(X) := g_m \circ \mathsf{Attn}_m \circ \cdots \circ g_1 \circ \mathsf{Attn}_1 \circ g_0(X) \quad \in \mathbb{F}p^{n \times d},$$

*where $\circ$ denotes the composition of functions.*

Additional building blocks of the tensor attention Transformers can be found in Appendix C.

## 3 COMPLEXITY OF TENSOR ATTENTION TRANSFORMER

We now formally turn our attention to investigating the circuit complexity of the tensor attention layer and the multi-layer tensor attention Transformer, emphasizing their computability within the complexity class $\mathsf{TC}^0$. Section 3.1 delves into matrix operations. Section 3.2 addresses the computation of a single tensor attention layer. Section 3.3 provides an in-depth examination of the entire tensor attention mechanism. Lastly, Section 3.4 presents our principal findings regarding the circuit complexity bounds for the tensor attention Transformer. These results establish the foundation for the main theorem concerning Transformer expressiveness.

### 3.1 MATRIX OPERATIONS

We demonstrate that fundamental matrix multiplication is efficiently evaluatable within $\mathsf{TC}^0$.

**Lemma 3.1** (Matrix multiplication in $\mathsf{TC}^0$, Lemma 4.2 in (Chen et al., 2024a)). *Let $A \in \mathbb{F}_p^{n_1 \times d}$, $B \in \mathbb{F}_p^{d \times n_2}$ represent matrices. Under conditions that $p \leq \mathrm{poly}(n)$, $n_1, n_2 \leq \mathrm{poly}(n)$, and $d \leq n$, the product $AB$ is evaluatable via $\mathrm{poly}(n)$ size uniform threshold circuit with $(d_{\mathrm{std}} + d_{\oplus})$ depth.*

We have similar conclusions for the Kronecker product.

**Lemma 3.2** (Kronecker product in $\mathsf{TC}^0$, informal version of Lemma E.1). *Let $A \in \mathbb{F}_p^{n_1 \times d}$ and $B \in \mathbb{F}_p^{d \times n_2}$ represent matrices. If $p \leq \mathrm{poly}(n)$, $n_1, n_2 \leq \mathrm{poly}(n)$, and $d \leq n$, the Kronecker product $A \otimes B$ can be evaluated by a $\mathrm{poly}(n)$ size uniform threshold circuit with $d_{\mathrm{std}}$ depth.*

**Lemma 3.3** (Column-wise Kronecker Product in $\mathsf{TC}^0$, informal version of Lemma E.2). *Let matrices $A \in \mathbb{F}_p^{n_1 \times d}$ and $B \in \mathbb{F}_p^{n_2 \times d}$ be given. If $p \leq \mathrm{poly}(n)$, $n_1, n_2 \leq \mathrm{poly}(n)$, and $d \leq n$, then the column-wise Kronecker product $A \oslash B$ is evaluatable by a $\mathrm{poly}(n)$ size uniform threshold circuit with depth $d_{\mathrm{std}}$.*

**Lemma 3.4** (Row-wise Kronecker Product Computation in $\mathsf{TC}^0$, informal version of Lemma E.3). *Let $A \in \mathbb{F}_p^{d \times n_1}$ and $B \in \mathbb{F}_p^{d \times n_2}$ be matrices, with the conditions $p \leq \mathrm{poly}(n)$, $n_1, n_2 \leq \mathrm{poly}(n)$, and $d \leq n$. Then, a size $\mathrm{poly}(n)$ uniform threshold circuit with $d_{\mathrm{std}}$ depth can calculate the row-wise Kronecker product $A \ominus B$.*

### 3.2 SINGLE TENSOR ATTENTION LAYER

Here, we examine the complexity of the single layer of the tensor attention.

**Lemma 3.5** (Complexity of Single Tensor Attention Layer in $\mathsf{TC}^0$, informal version of Lemma E.4). *When $p \leq \mathrm{poly}(n)$, the $\mathsf{Attn}$ in Definition 2.20, is evaluatable by a $\mathrm{poly}(n)$ size and $5d_{\mathrm{std}} + 5d_{\oplus} + d_{\exp}$ depth uniform threshold circuit.*

### 3.3 MULTI-LAYER TENSOR ATTENTION

This section analyzes the computation of multi-layer tensor attention in a Transformer.

**Lemma 3.6** (Computation of Multi-layer Tensor Attention Transformer in $\mathsf{TC}^0$, informal version of Lemma E.5). *Suppose that for every $i \in [m]$, the function $g_i$ in $\mathsf{TF}$ can be evaluated by a $\mathrm{poly}(n)$ size constant depth $d_g$ uniform threshold circuit. Assuming that $p \leq \mathrm{poly}(n)$, the $\mathsf{RoPE}$-based tensor attention $\mathsf{TF}$, as defined in Definition 2.21, is evaluatable by $\mathrm{poly}(n)$ size uniform threshold circuit of and depth $(m + 1)d_g + 6md_{\mathrm{std}} + 5md_{\oplus} + md_{\exp}$.*

### 3.4 Circuit Complexity Bound of Tensor Attention

The subsequent discussion focuses on presenting the main result regarding the circuit complexity bound for tensor attention Transformers.

**Theorem 3.7** (Circuit Complexity of Tensor Attention, informal version of Theorem E.6). *Assume that for every $i \in [m]$, the function $g_i$ in TF is evaluatable by $\mathrm{poly}(n)$ size uniform threshold circuit of constant $d_g$ depth. As Definition 2.21, we can approximate the RoPE-based tensor attention Transformer TF by a uniform $\mathsf{TC}^0$ circuit family, when $d \leq O(n)$, $p \leq \mathrm{poly}(n)$, and $m \leq O(1)$.*

Above Theorem E.6, we establish that, unless $\mathsf{TC}^0 = \mathsf{NC}^1$, a constant depth tensor attention with $\mathrm{poly}(n)$ size, and $\mathrm{poly}(n)$-precision can be approximated by a DLOGTIME-uniform $\mathsf{TC}^0$ circuit family. While tensor attention Transformers exhibit strong empirical performance, this result indicates inherent limits in their expressivity when viewed through the framework of circuit complexity. These constraints are examined further in Section 5, in tandem with the analysis from Section 4.

## 4 Complexity of RoPE-based Tensor Attention Transformer

This section presents key results concerning the circuit complexity of fundamental operations within RoPE-based tensor attention computations. Section 4.1 investigates trigonometric functions, which play a crucial role in rotary position embeddings, while Section 4.2 focuses on the RoPE-based tensor attention matrix computation. Section 4.3 delves into the individual RoPE-based tensor attention layer. In Section 4.4, the complete RoPE-based tensor attention mechanism is detailed. Finally, Section 4.5 presents the primary results regarding the circuit complexity bounds for RoPE-based tensor attention, forming the foundation for the essential theorem on RoPE-based Tensor Attention Transformer expressiveness.

### 4.1 Approximating Trigonometric Functions

Here, we outline the efficient calculation of fundamental trigonometric functions that are critical for RoPE embeddings via threshold circuits. The next lemma plays a central role:

**Lemma 4.1** (Trigonometric Function Approximation in $\mathsf{TC}^0$, Lemma 4.1 in (Chen et al., 2024a)). *For any $p \leq \mathrm{poly}(n)$, the values of $\sin(x)$ and $\cos(x)$ for a float point number $x$ of $p$ bits with a relative error bounded by $2^{-p}$ are evaluatable by $\mathrm{poly}(n)$ size uniform threshold circuit with constant depth. Let $d_\triangle$ denote the maximum depth required to calculate both $\cos(x)$ and $\sin(x)$.*

### 4.2 RoPE-based Tensor Attention Matrix

The following section builds on what we already know about the computation of the RoPE-based tensor attention matrix.

**Lemma 4.2** (RoPE-based tensor attention matrix computation in $\mathsf{TC}^0$, informal version of Lemma E.7). *For any polynomial $p \leq \mathrm{poly}(n)$, a size $\mathrm{poly}(n)$ uniform threshold circuit with depth $7d_{\mathrm{std}} + 4d_\oplus + d_\triangle + d_{\exp}$ is capable of computing $A$, i.e., the attention matrix in Definition 2.17.*

### 4.3 Single RoPE-based Tensor Attention Layer

This section provides a detailed examination of the RoPE tensor attention layer, with an emphasis on tracking the circuit depth requirements throughout the computation process.

**Lemma 4.3** (One RoPE-based Attention Layer within $\mathsf{TC}^0$, informal version of Lemma E.8). *For $p \leq \mathrm{poly}(n)$, the Attn defined in Definition 2.18 can be evaluatable by a $\mathrm{poly}(n)$ depth $11d_{\mathrm{std}} + 8d_\oplus + d_\triangle + d_{\exp}$ uniform threshold circuit.*

### 4.4 Multi-layer RoPE Tensor Attention

We now describe the computation of the multi-layer RoPE-based tensor attention Transformer.

**Lemma 4.4** (Multi-layer RoPE-based tensor attention Transformer computation in $\mathsf{TC}^0$, informal version of Lemma E.9). *Consider the assumption that for every $i \in [m]$, $g_i$ in TF can be evaluated*

*using* $\mathrm{poly}(n)$ *size uniform threshold circuit with a constant depth* $d_g$. *When* $p \leq \mathrm{poly}(n)$, *the* RoPE-*based tensor attention* TF, *as specified in Definition 2.21, can be evaluated by* $\mathrm{poly}(n)$ *size uniform threshold circuit of depth* $(m+1)d_g + 11md_{\mathrm{std}} + 8md_{\oplus} + m(d_{\triangle} + d_{\exp})$.

## 4.5 Main Result: Circuit Complexity of RoPE Tensor Attention

We present the central contribution of this paper, establishing the circuit complexity for the RoPE-based tensor attention.

**Theorem 4.5** (Main result, Circuit complexity of RoPE-based tensor attention Transformers, informal version of Theorem E.10). *Assume that* $\forall i \in [m]$, $g_i$ *in* TF *can be computed using* $\mathrm{poly}(n)$ *size uniform threshold circuit of constant depth* $d_g$. *The* RoPE-*based tensor attention* TF, *as defined in Definition 2.21, is simulatable by uniform* $\mathsf{TC}^0$ *circuit family when* $d \leq O(n), p \leq \mathrm{poly}(n)$, *and* $m \leq O(1)$.

In Theorem 3.7 and Theorem 4.5, unless $\mathsf{TC}^0 = \mathsf{NC}^1$, a DLOGTIME-uniform $\mathsf{TC}^0$ circuit family can emulate both tensor attention Transformers and RoPE-based tensor attention Transformers, which are defined by constant depth, $\mathrm{poly}(n)$ precision, and $\mathrm{poly}(n)$ size. This finding suggests that, notwithstanding the empirical success of these models, their expressive capabilities are intrinsically constrained when analyzed through the lens of circuit complexity. The subsequent section will delve deeper into these limitations.

**Difference to (Chiang, 2024).** Our work studies a fundamentally different Transformer architecture from (Chiang, 2024). While (Chiang, 2024) analyzes the circuit complexity of average-hard attention and standard softmax attention, we focus on tensor attention Transformers, a distinct and more expressive architecture. Our main results establish that tensor attention (both without RoPE and with RoPE) lies in $\mathsf{TC}^0$ (Theorems 3.7 and 4.5), and we further identify several $\mathsf{NC}^1$-hard problems that tensor attention cannot solve (Theorems 5.9 and 5.10).

Technically, our contributions differ from (Chiang, 2024) in two important ways. First, to show that tensor attention is in $\mathsf{TC}^0$, we must analyze the circuit complexity of tensor operations such as the Kronecker product, as well as its row-wise and column-wise variants inside the tensor attention mechanism. These results are established in Lemmas 3.5–3.6 and are different from (Chiang, 2024). Second, to analyze the computability of the RoPE rotation matrices within $\mathsf{TC}^0$, we rely on a recent number-theoretic tool (Lemma 4.1), which is also not used in (Chiang, 2024).

## 5 Hardness

This section delineates two fundamental problems, accompanied by their respective hardness results. The fixed membership problem is introduced in Section 5.1, while the closure problem is defined in Section 5.2. Section 5.3 presents the four principal hardness results.

### 5.1 Fixed Membership

The fixed membership problem, as originally formulated in (Fleischer & Kufleitner, 2019), is thoroughly defined in this section. Before introducing this important problem with close connections to weak recognition in automata theory (Perrin & Pin, 2004; Fleischer & Kufleitner, 2016; 2019), we first introduce some basic notations and concepts.

**Definition 5.1** (Infinite iteration, implicit in page 3 of (Fleischer & Kufleitner, 2016)). *Let* $L$ *be a finite language and* $\epsilon$ *denote the empty string. We define the infinite iteration as* $L^{\omega} := \{u_1 u_2 u_3 \cdots \mid u_i \in L\}$ *if* $\epsilon \notin L$, *and define the infinite iteration as* $\{u_1 u_2 u_3 \cdots \mid u_i \in L \setminus \{\epsilon\}\} \cup \{\epsilon\}$ *otherwise.*

**Definition 5.2** (Kleene plus, implicit in page 2 of (Fleischer & Kufleitner, 2016)). *Let* $L$ *be a language. The Kleene plus of* $L$, *denoted by* $L^+$, *is the set of all finite concatenations of strings from* $L$, *i.e.,* $L^+ = \bigcup_{n \geq 1} L^n$, *where* $L^n$ *is the* $n$-*fold concatenation of* $L$.

**Definition 5.3** (Fixed membership problem, Definition from (Fleischer & Kufleitner, 2019)). *Let* $A$ *be a finite alphabet. Let* $h : A^+ \to S$ *be a fixed morphism into a finite semigroup* $S$. *Let* $F(S)$ *denote the collection of finite subsets of* $S$. *Given two finite words* $u, v \in A^+$ *and a fixed set* $P \subseteq F(S)$, *the fixed membership problem is the decision problem that asks the following: is* $uv^{\omega} \in h^{-1}(P)$?

Now, we present a formal exposition of the definition of the fixed membership problem, and then explain this problem with a concrete example.

**Example 5.4** (An example for fixed membership problem). *Let $A = \{\mathsf{a}, \mathsf{b}\}$ and let the finite semi-group be $S = \{0, 1\}$ with the operation defined by $x \cdot y = \max\{x, y\}$. We define a morphism $h : A^+ \to S$ by $h(\mathsf{a}) = 1, h(\mathsf{b}) = 0$ and extend $h$ to words by concatenation, i.e.,*

$$h(a_1 a_2 \cdots a_n) = h(x_1) h(x_2) \cdots h(x_n), \quad \forall a_1 a_2 \cdots a_n \in A^+$$

*In this example, we choose the fixed set to be $P = \{\{1\}\} \subseteq F(S)$, and choose $u = \mathsf{b}, v = \mathsf{a}$.*

*Then, we have $uv^\omega = \mathsf{baaa}\cdots$ and $h(uv^\omega) = 1$. Since $\{1\} \in P$, we conclude that $uv^\omega \in h^{-1}(P)$.*

*Thus, in this instance, the fixed membership problem returns YES.*

**Proposition 5.5** (Proposition 7.1 from (Fleischer & Kufleitner, 2019)). *The fixed membership problem for recognizing morphisms over finite words is $\mathsf{NC}^1$-complete.*

### 5.2 $(A_{F,r})^*$ CLOSURE

In this section, attention is shifted to the $(A_{F,r})^*$ closure problem in (Allender et al., 2003).

**Definition 5.6** (Kleene star, Definition 7.1 from (Allender et al., 2003)). *Let $L$ be a language and $\epsilon$ denotes the empty string. The Kleene star of $L$, denoted by $L^*$, is the set of all finite concatenations of strings from $L$: $L^* = \bigcup_{n \geq 0} L^n$, where $L^0 := \{\epsilon\}$ and $L^n$ denotes the $n$-fold concatenation of $L$.*

**Definition 5.7** ($(A_{F,r})^*$ Closure Problem, Definition 7.1 from (Allender et al., 2003)). *Let $(A, \circ)$ denote a finite monoid. A natural homomorphism $v : A^* \to A$ maps each word $w$ to its corresponding valuation $v(w)$ in the monoid $A$. Let $F \subseteq A$ and $r \in \mathbb{Z}_+$. For a word $w$, we write $|w|$ for its length. The language $A_{F,r} \subseteq A^*$ is defined as*

$$A_{F,r} = \{w \in A^* \mid |w| \leq r, v(w) \in F\}.$$

*The $(A_{F,r})^*$ closure problem refers to the decision problem aimed at determining whether a given string $s$ belongs to $(A_{F,r})^*$.*

We now introduce a famous result from previous work, which will be used later.

**Theorem 5.8** (Theorem 7.3(a) from (Allender et al., 2003)). *For any nonsolvable monoid $A$, there exists a group $F \subseteq A$ and a constant $r > 0$ such that the $(A_{F,r})^*$ closure problem is $\mathsf{NC}^1$-complete.*

### 5.3 HARDNESS RESULT

We present two crucial findings concerning tensor attention Transformers and RoPE-based tensor attention Transformers.

**Theorem 5.9** (Informal version of Theorem F.1). *If $\mathsf{TC}^0 \neq \mathsf{NC}^1$, an $O(1)$ layers RoPE-based tensor attention Transformer with $d \leq O(n)$ hidden dimension, $\mathrm{poly}(n)$ precision is incapable of solving the fixed membership problem.*

**Theorem 5.10** (Informal version of Theorem F.2). *Assuming $\mathsf{TC}^0 \neq \mathsf{NC}^1$, an $O(1)$ layers RoPE-tensor attention Transformer with $d \leq O(n)$ hidden dimension, and $\mathrm{poly}(n)$ precision is not capable of solving the $(A_{F,r})^*$ closure problem.*

The above two theorems show the representation limitation of a RoPE-based tensor attention Transformer with a constant number of layers.

## 6 CONCLUSION

This paper analyzes the computational limits of tensor attention Transformers and extends to its RoPE-based variants, showing they are simulable by uniform $\mathsf{TC}^0$ circuits and, under $\mathsf{TC}^0 \neq \mathsf{NC}^1$, cannot solve fixed membership or $(A_{F,r})^*$ closure problems with $O(1)$ layers, $\mathrm{poly}(n)$ precision, and $d \leq O(n)$ dimensions. Despite their empirical success, these models face fundamental trade-offs between efficiency and expressive power. The analysis, limited to constant-depth activations, invites further research into alternative attention mechanisms and encoding schemes to bridge the gap between theoretical constraints and practical performance.

## ETHIC STATEMENT

This paper does not involve human subjects, personally identifiable data, or sensitive applications. We do not foresee direct ethical risks. We follow the ICLR Code of Ethics and affirm that all aspects of this research comply with the principles of fairness, transparency, and integrity.

## REPRODUCIBILITY STATEMENT

We ensure reproducibility of our theoretical results by including all formal assumptions, definitions, and complete proofs in the appendix. The main text states each theorem clearly and refers to the detailed proofs. No external data or software is required.

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

# Appendix

## A RELATED WORK

**The Computational Complexity in Deep Learning.** Circuit complexity studies computational models using circuit families, with classes like $\mathsf{AC}^0$ and $\mathsf{TC}^0$ characterizing problems solvable by parallel circuits with logic or threshold gates, respectively, and $\mathsf{NC}^1$ solving problems with $O(\log n)$ depth (Merrill et al., 2022). It is known that $\mathsf{AC}^0 \subset \mathsf{TC}^0 \subseteq \mathsf{NC}^1$, though whether $\mathsf{TC}^0 \neq \mathsf{NC}^1$ remains open. Assuming this inequality, (Liu et al., 2022) shows that Transformer depth must scale with input length for simulating certain non-solvable semiautomata. Circuit complexity also evaluates architectures like Mamba (Chen et al., 2024b) and Hopfield networks (Li et al., 2024).

**Computation of Transformers.** Transformers have revolutionized natural language processing but struggle with mathematical computations (Charton, 2022). Research has focused on their computational limits, particularly for two types: (1) average-head attention Transformers, which set the highest probability to 1 and others to 0, and (2) softmax-attention Transformers, which use the softmax function. Merrill, Sabharwal, and Smith (Merrill et al., 2022) show that average-head attention Transformers exceed $\mathsf{AC}^0$ power but are simulable by constant-depth threshold circuits in the non-uniform $\mathsf{TC}^0$ class. Similarly, (Liu et al., 2022) show softmax-attention Transformers also belong to $\mathsf{TC}^0$. Further studies (Merrill & Sabharwal, 2023; 2024) refine these results, demonstrating that these models fit within L-uniform and DLOGTIME-uniform $\mathsf{TC}^0$ classes. In practical applications, (Feng et al., 2024) argue that unless $\mathsf{TC}^0 = \mathsf{NC}^1$, Transformers with log-precision cannot efficiently solve arithmetic or CFG membership problems (Sipser, 1996), highlighting their limitations in math tasks.

**Tensor Computation for High-order Representation.** Tensors are more effective than matrices in capturing higher-order relationships in data and are essential for low-rank factorizations in various fields, including natural language processing (Lei et al., 2015; Bouchard et al., 2015), computer vision (Lu et al., 2016; Chen et al., 2017), computer graphics (Wang et al., 2005; Vasilescu, 2009), security (Acar et al., 2006; Kolda & Bader, 2006), and data mining (Karatzoglou et al., 2010; Rendle & Schmidt-Thieme, 2010; Mørup, 2011). Tensors are also crucial in machine learning (Podosinnikova et al., 2015; Jain & Oh, 2014; Zhong et al., 2017; Yang et al., 2019; Shi et al., 2022) and other domains (Reps et al., 2016; Yi et al., 2016; Ray et al., 2018).

**Roadmap.** In Section B, we introduce four fundamental float point operations used in this paper. Section C mainly discussing two widely used components when constructing Transformer. In Section C, the complete proof of Fact D.1 has been proposed. And in Section E, all the missing proofs from Section 3 and Section 4 are completed. In Section F, we consummate all the missing proofs appear in Section 5.

## B FLOATING-POINT NUMBER OPERATIONS

**Definition B.1** (Float point operations, Lemma 10 from (Chiang, 2024)). *Let $x$ and $y$ represent two integers, then $x \oslash y$ defined as follows:*

$$x \oslash y := \begin{cases} 1/8 + x/y & \text{if } x/y \text{ is not a multiple of } 1/4, \\ x/y & \text{if } x/y \text{ is a multiple of } 1/4. \end{cases}$$

*Let $\langle r_1, k_1 \rangle$ and $\langle r_2, k_2 \rangle$ all denoted as $p$-bit float points, then we have:*

- *Addition:*

$$\langle r_1, k_1 \rangle + \langle r_2, k_2 \rangle$$
$$:= \begin{cases} \text{round}_p(\langle r_1 + r_2 \oslash 2^{k_1 - k_2}, k_1 \rangle) & \text{if } k_1 \geq k_2, \\ \text{round}_p(\langle r_1 \oslash 2^{k_2 - k_1} + r_2, k_2 \rangle) & \text{if } k_1 \leq k_2. \end{cases}$$

- **Comparison:**

$$\langle r_1, k_1 \rangle \leq \langle r_2, k_2 \rangle \Leftrightarrow \begin{cases} r_1 \leq r_2 \oslash 2^{k_1 - k_2} & \text{if } k_1 \geq k_2, \\ r_1 \oslash 2^{k_2 - k_1} \leq r_2 & \text{if } k_1 \leq k_2. \end{cases}$$

- **Multiplication:**

$$\langle r_1, k_1 \rangle \times \langle r_2, k_2 \rangle := \text{round}_p(\langle r_1 r_2, k_1 + k_2 \rangle).$$

- **Division:**

$$\langle r_1, k_1 \rangle \div \langle r_2, k_2 \rangle$$
$$:= \text{round}_p(\langle r_1 2^{p-1} \oslash r_2, k_1 - k_2 - p + 1 \rangle).$$

- **Floor:**

$$\lfloor \langle r, k \rangle \rfloor := \begin{cases} \text{round}(\langle r/2^{-k}, 0 \rangle) & \text{if } k < 0, \\ \langle r 2^k, 0 \rangle & \text{if } k \geq 0. \end{cases}$$

The operations mentioned above are capable of efficient hardware implementation, as demonstrated by the following:

**Lemma B.2** (Float point operations in $\mathsf{TC}^0$, Lemma 10 and Lemma 11 from (Chiang, 2024)). *If integer $0 < p \leq \text{poly}(n)$, then we say the conditions below are satisfied:*

- **Part 1.** *The operations addition, division, multiplication, and comparison of two $p$-bit float point numbers (described in Definition B.1) are calculable by a constant depth $\text{poly}(n)$ size uniform threshold circuit. $d_{\text{std}}$ denotes the deepest depth necessitated for executing these operations.*

- **Part 2.** *We can execute $n$ $p$-bit float point numbers repeated multiplication using a constant depth $\text{poly}(n)$ size uniform threshold circuit. The required depth for this iterated multiplication process is denoted as $d_{\otimes}$.*

- **Part 3.** *We can approximate $n$ $p$-bit float point numbers sequential addition and rounding using a constant depth $\text{poly}(n)$ size uniform threshold circuit. The depth needed for iterated addition is represented by $d_{\oplus}$.*

**Corollary B.3** (Floor operation in $\mathsf{TC}^0$, Corollary 3.17 from (Chen et al., 2024a)). *For any integer $0 < p \leq \text{poly}(n)$, a $\text{poly}(n)$ size constant depth uniform threshold circuit is able to calculate the floor operation from Definition B.1 on a $p$-bit float point number. The operation's maximum depth is bounded by $d_{\text{std}}$, as established in Lemma B.2.*

**Lemma B.4** (Computing $\exp$ in $\mathsf{TC}^0$, Lemma 12 from (Chiang, 2024)). *For any integer $0 < p \leq \text{poly}(n)$ and any $p$-bit float point number $x$, it is computable to approximate most $2^{-p}$ relative error $\exp(x)$ using $\text{poly}(n)$ size constant depth uniform threshold circuit. The depth required for this computation is denoted by $d_{\exp}$.*

**Lemma B.5** (Computing square root in $\mathsf{TC}^0$, Lemma 12 from (Chiang, 2024)). *Given an integer $p$ such that $0 < p \leq \text{poly}(n)$ and a $p$-bit float point number $x$, a constant depth $\text{poly}(n)$ size uniform threshold circuit exists to calculate $\sqrt{x}$ with a relative error bounded by $2^{-p}$. The depth required for this operation is represented by $d_{\text{sqrt}}$.*

## C  OTHER BUILDING BLOCKS OF TRANSFORMERS

We begin by introducing two basic building blocks: softmax functions and the rotation matrix.

**Definition C.1** (Softmax function). *Noted $z \in \mathbb{F}_p^n$. The $\mathsf{Softmax}$ function : $\mathbb{F}_p^n \to \mathbb{F}_p^n$ is formally given by:* $\mathsf{Softmax}(z) := \exp(z)/\langle \exp(z), \mathbf{1}_n \rangle$.

One of the pivotal advancements in contemporary Transformer architectures is RoPE, which employs a rotation matrix as its foundation:

**Definition C.2** (Rotation matrix block). *For an input sequence of length $n$, embedding dimension $d$, and parameter $\theta \in \mathbb{F}_p$, the rotation matrix is constructed as follows:*

$$R(\theta) := \begin{bmatrix} \cos\theta & -\sin\theta \\ \sin\theta & \cos\theta \end{bmatrix}.$$

Subsequently, we define two categories of $g_i$ functions. Start with the layer normalization.

**Definition C.3** (Layer normalization). *Let $X \in \mathbb{F}_p^{n \times d}$ be the input data matrix, and let $i \in [n]$. The LN layer is formulated as:*

$$g^{\mathrm{LN}}(X)_{i,*} := \frac{X_{i,*} - \mu_i}{\sqrt{\sigma_i^2}},$$

*where $\mu_i := \sum_{j=1}^d \frac{X_{i,j}}{d}$, and $\sigma_i^2 := \sum_{j=1}^d \frac{(X_{i,j} - \mu_i)^2}{d}$.*

The second category is the multilayer perceptron.

**Definition C.4** (Multilayer perceptron). *Let $X \in \mathbb{F}_p^{n \times d}$ be the input data matrix, and let $i \in [n]$. The MLP layer is described as:*

$$g^{\mathrm{MLP}}(X)_{i,*} := \underbrace{W}_{d \times d} \cdot \underbrace{X_{i,*}}_{d \times 1} + \underbrace{b}_{d \times 1}.$$

The foundation of modern Transformer is built upon these layered architectures, which integrate float point computations, attention, and rotation matrix to an exceptionally efficient framework for sequential computation.

According to Definition 2.21, the definition of the Multi-layer RoPE-based Transformer is provided, which integrates RoPE-based self-attention layers together with supplementary components, such as layer normalization and MLP. This section subsequently addresses the circuit complexity associated with these mechanisms.

The analysis begins with an investigation of the complexity pertaining to the MLP layer.

**Lemma C.5** (Compute MLP in $\mathsf{TC}^0$, Lemma 4.5 in (Chen et al., 2024a)). *If $p \leq \mathrm{poly}(n)$, $\mathrm{poly}(n)$ size depth $2d_{\mathrm{std}} + d_{\oplus}$ uniform threshold circuit suffices to evaluate the MLP layer as defined in Definition C.4.*

Next, we will turn our attention to the complexity of the LN layer.

**Lemma C.6** (Compute Layer-norm in $\mathsf{TC}^0$, Lemma 4.6 in (Chen et al., 2024a)). *Let $p \leq \mathrm{poly}(n)$. Then, the layer-normalization defined in Definition C.3 can be evaluated by $\mathrm{poly}(n)$ size depth $5d_{\mathrm{std}} + 2d_{\oplus} + d_{\mathrm{sqrt}}$ uniform threshold circuit.*

## D  SWAP RULE OF TENSOR AND MATRIX PRODUCT

In this section, we present a useful fact, which indicates that the order of tensor operation and matrix multiplication can be swapped, enabling computation in the lower dimension first to reduce complexity.

**Fact D.1** (Swap rule for tensor and matrix product). *Let $W_1, W_2 \in \mathbb{R}^{d \times d}$, $A_1, A_2 \in \mathbb{R}^{n \times d}$. We have*

$$\underbrace{(A_1 \otimes A_2)}_{n^2 \times d^2} \cdot \underbrace{(W_1 \oslash W_2)}_{d^2 \times d} = \underbrace{(A_1 \cdot W_1)}_{n \times d} \oslash \underbrace{(A_2 \cdot W_2)}_{n \times d}.$$

*Proof.* For any $i_1, i_2 \in [n], j \in [d]$, we have

$$((A_1 \otimes A_2) \cdot (W_1 \oslash W_2))_{i_1 + (i_2 - 1)n, j}$$
$$= \sum_{k_1 \in [d], k_2 \in [d]} (A_1 \otimes A_2)_{i_1 + (i_2 - 1)n, k_1 + (k_2 - 1)d}$$

$$\cdot (W_1 \oslash W_2)_{k_1+(k_2-1)d,j}$$
$$= \sum_{k_1 \in [d], k_2 \in [d]} (A_1 \otimes A_2)_{i_1+(i_2-1)n, k_1+(k_2-1)d}$$
$$\cdot (W_1)_{k_1,j} \cdot (W_2)_{k_2,j}$$
$$= \sum_{k_1 \in [d], k_2 \in [d]} (A_1)_{i_1,k_1} \cdot (A_2)_{i_2,k_2} \cdot (W_1)_{k_1,j} \cdot (W_2)_{k_2,j}$$
$$= \left( \sum_{k_1 \in [d]} (A_1)_{i_1,k_1} \cdot (W_1)_{k_1,j} \right) \cdot \left( \sum_{k_2 \in [d]} (A_2)_{i_2,k_2} \cdot (W_2)_{k_2,j} \right)$$
$$= (A_1 \cdot W_1)_{i_1,j} \cdot (A_2 \cdot W_2)_{i_2,j}$$
$$= ((A_1 \cdot W_1) \oslash (A_2 \cdot W_2))_{i_1+(i_2-1)n,j},$$

where the initial step involves the application of matrix multiplication, followed by the utilization of Definition 2.13 in the second step. Subsequently, the third step employs Definition 2.12, while the fourth step simplifies the expression through fundamental algebraic principles. The fifth step re-engages matrix multiplication, and the concluding step leverages Definition 2.13 once more. □

## E  MSSING PROOFS IN SECTION 3 AND SECTION 4

Here we present some missing proofs in Section 3 and Section 4. First we show the proof of Lemma E.1 below.

**Lemma E.1** (Formal version of Lemma 3.2). *Let $A \in \mathbb{F}_p^{n_1 \times d}$ and $B \in \mathbb{F}_p^{d \times n_2}$ represent matrices. If $p \leq \mathrm{poly}(n)$, $n_1, n_2 \leq \mathrm{poly}(n)$, and $d \leq n$, the Kronecker product $A \otimes B$ can be evaluated by a $\mathrm{poly}(n)$ size uniform threshold circuit with $d_{\mathrm{std}}$ depth.*

*Proof.* Each product $(A)_{i_1,j_1} \cdot (B)_{i_2,j_2}$ computes the entry $(A \otimes B)_{i_1+(i_2-1)n_1, j_1+(j_2-1)d}$, according to Part 1 of Lemma B.2. Since the computations for distinct index pairs $(i_1, j_1)$ and $(i_2, j_2)$ are independent, they can be performed concurrently, resulting in a total depth of $d_{\mathrm{std}}$ for all computations.

The circuit size is polynomial in $n$, as each operation uses a polynomial-sized circuit, and $n_1, n_2, d \leq \mathrm{poly}(n)$.

Therefore, the Kronecker product $A \otimes B$ is evaluatable by a $\mathrm{poly}(n)$ size uniform threshold circuit with $d_{\mathrm{std}}$ depth.

This concludes the proof. □

**Lemma E.2** (Formal version of Lemma 3.3). *Let matrices $A \in \mathbb{F}_p^{n_1 \times d}$ and $B \in \mathbb{F}_p^{n_2 \times d}$ be given. If $p \leq \mathrm{poly}(n)$, $n_1, n_2 \leq \mathrm{poly}(n)$, and $d \leq n$, then the column-wise Kronecker product $A \oslash B$ is evaluatable by a $\mathrm{poly}(n)$ size uniform threshold circuit with depth $d_{\mathrm{std}}$.*

*Proof.* This result directly follows from Lemma E.1. By applying Lemma B.2, the product $(A)_{i_1,j} \cdot (B)_{i_2,j}$ for $i_1 \in [n_1]$, $i_2 \in [n_2]$, and $j \in [d]$ computes the entry $(A \oslash B)_{i_1+(i_2-1)n_1,j}$ using a uniform threshold circuit with depth $d_{\mathrm{std}}$. Since these computations are independent for distinct values of $(i_1, i_2)$, they can be evaluated concurrently, resulting in a circuit depth of $d_{\mathrm{std}}$.

The circuit size remains polynomial in $n$ because $n_1, n_2, d \leq \mathrm{poly}(n)$ and every operation utilizes a polynomial-sized circuit.

Thus, the column-wise Kronecker product $A \oslash B$ can be evaluated by a $\mathrm{poly}(n)$ size depth $d_{\mathrm{std}}$ uniform threshold circuit.

This concludes the proof. □

**Lemma E.3** (Formal version of Lemma 3.4). *Let $A \in \mathbb{F}_p^{d \times n_1}$ and $B \in \mathbb{F}_p^{d \times n_2}$ be matrices, with the conditions $p \leq \mathrm{poly}(n)$, $n_1, n_2 \leq \mathrm{poly}(n)$, and $d \leq n$. Then, a size $\mathrm{poly}(n)$ uniform threshold circuit with $d_{\mathrm{std}}$ depth can calculate the row-wise Kronecker product $A \ominus B$.*

*Proof.* Similarly as Lemma E.2, according to Lemma B.2, the product $(A)_{i,j_1} \cdot (B)_{i,j_2}$, for $j_1 \in [n_1]$, $j_2 \in [n_2]$, and $i \in [d]$, computes the entry $(A \oslash B)_{i,j_1+(j_2-1)n_1}$ via a depth $d_{\mathrm{std}}$ uniform threshold circuit. These products, for distinct $(i_1, i_2)$, are evaluatable in parallel, allowing all necessary products $(A)_{i,j_1} \cdot (B)_{i,j_2}$ to be evaluated simultaneously within the depth $d_{\mathrm{std}}$.

The circuit size is polynomial in $n$ because $n_1, n_2, d \leq \mathrm{poly}(n)$, and each individual operation can be evaluated by a polynomial-sized circuit.

Hence, $\mathrm{poly}(n)$ size $d_{\mathrm{std}}$ depth uniform threshold circuit can calculate $A \oslash B$.

The proof is concluded. $\qquad\square$

**Lemma E.4** (Formal version of Lemma 3.5). *When $p \leq \mathrm{poly}(n)$, the attention* Attn *in Definition 2.20, is evaluatable by a* $\mathrm{poly}(n)$ *size and* $5d_{\mathrm{std}} + 5d_{\oplus} + d_{\exp}$ *depth uniform threshold circuit.*

*Proof.* The matrix multiplications $Q := ZW_Q$, $K_1 := ZW_{K_1}$, and $K_2 := ZW_{K_2}$ can be evaluated in parallel with a size $\mathrm{poly}(n)$ depth $d_{\mathrm{std}} + d_{\oplus}$ uniform threshold circuit, as established in Lemma 3.1.

As per Lemma E.2, the column-wise Kronecker product $V := V_1 \oslash V_2$ is evaluatable for $\mathrm{poly}(n)$ size uniform threshold circuit with $d_{\mathrm{std}}$ depth.

Using Lemma 3.1 and Part 1 of Lemma B.3, the operation $QK^{\top}/d$ is evaluatable by $\mathrm{poly}(n)$ size uniform threshold circuit with depth $2d_{\mathrm{std}} + d_{\oplus}$.

According to Lemma B.4, the exponential function $exp()$ is evaluatable by $\mathrm{poly}(n)$ size uniform threshold circuit with depth $d_{\exp}$.

As per Part 3 of Lemma B.2, $D := A\mathbf{1}n$ is evaluated with $\mathrm{poly}(n)$ size uniform threshold circuit of depth $d\oplus$.

Finally, the expression $D^{-1}AV$ is evaluated in parallel using $\mathrm{poly}(n)$ size uniform threshold circuit of $2(d_{\mathrm{std}} + d_{\oplus})$ depth, as shown in Lemma 3.1.

The total depth required for computing $\mathsf{Attn}_i(X) := D^{-1}AV$ is therefore:

$$6d_{\mathrm{std}} + 5d_{\oplus} + d_{\exp}.$$

$\qquad\square$

**Lemma E.5** (Formal version of Lemma 3.6). *Suppose that for every $i \in [m]$, the function $g_i$ in* TF *can be evaluated by a* $\mathrm{poly}(n)$ *size constant depth $d_g$ uniform threshold circuit. Assuming that $p \leq \mathrm{poly}(n)$, the* RoPE-*based tensor attention* TF*, as defined in Definition 2.21, is evaluatable by* $\mathrm{poly}(n)$ *size uniform threshold circuit of and depth $(m+1)d_g + 6md_{\mathrm{std}} + 5md_{\oplus} + md_{\exp}$.*

*Proof.* By assumption, $\forall i \in [m]$, $g_i$ is evaluatable by a $\mathrm{poly}(n)$ size constant $d_g$ depth uniform threshold circuit. From Lemma E.4, the attention operation $\mathsf{Attn}_i$ is evaluatable by $\mathrm{poly}(n)$ size uniform threshold circuit with depth $6d_{\mathrm{std}} + 5d_{\oplus} + d_{\exp}$.

In order to compute $\mathsf{TF}(X)$, the functions $g_0, g_1, \ldots, g_m$ and $\mathsf{Attn}_1, \ldots, \mathsf{Attn}_m$ must be evaluated. Consequently, the overall depth is $(m+1)d_g + 6md_{\mathrm{std}} + 5md_{\oplus} + md_{\exp}$, and the circuit size remains $\mathrm{poly}(n)$.

The proof is completed. $\qquad\square$

**Theorem E.6** (Formal version of Theorem 3.7). *Assume that for every $i \in [m]$, the function $g_i$ in* TF *is evaluatable by* $\mathrm{poly}(n)$ *size uniform threshold circuit of constant $d_g$ depth. As described in Definition 2.21, we can approximate the* RoPE-*based tensor attention Transformer* TF *by a uniform* $\mathsf{TC}^0$ *circuit family, when $d \leq O(n)$, $p \leq \mathrm{poly}(n)$, and $m \leq O(1)$.*

*Proof.* With constant $m$, and Lemma E.5, the depth of the circuit computing $\mathsf{TF}(X)$ is

$$(m+1)d_g + 6md_{\mathrm{std}} + 5md_{\oplus} + md_{\exp} = O(1),$$

and the $\text{poly}(n)$ circuit size. Thus, a uniform $\mathsf{TC}^0$ circuit family can simulate this computation.

This concludes the proof. $\square$

**Lemma E.7** (Formal version of Lemma 4.2). *For any polynomial $p \leq \text{poly}(n)$, a size $\text{poly}(n)$ uniform threshold circuit with depth $7d_{\text{std}} + 4d_{\oplus} + d_{\triangle} + d_{\exp}$ is capable of computing A, i.e., the attention matrix in Definition 2.17.*

*Proof.* For every $j_1, j_2, j_3 \in [n]$, the matrix element $A_{j_1, j_2+(j_3-1)d}$ is evaluated according to the formula in Definition 2.17.

From Lemma 3.1, the matrix products $Q := ZW_Q$, $K_1 := ZW_{K_1}$, and $K_2 := ZW_{K_2}$ can be evaluated in parallel by a size $\text{poly}\, n$ depth $d_{\text{std}} + d_{\oplus}$ uniform threshold circuit.

As indicated by Lemma 4.1, the entries of $R_{j_1-j_2}$ are evaluatable by a size $\text{poly}(n)$ depth $d_{\triangle}$ uniform threshold circuit. Since $n$ is polynomial, all entries of $R_{j_1-j_2}$ are evaluatable simultaneously with the same circuit size and depth. This holds true for $R_{j_1-j_3}$ and $R_{j_1-j_2}$ as well.

According to Lemma E.3, the row-wise Kronecker product $R_{j_1,j_2+(j_3-1)d} = R_{j_1-j_2} \ominus R_{j_1-j_3}$ is evaluatable by $\text{poly}\, n$ size uniform threshold circuit with $d_{\text{std}}$ depth.

Lemma E.1 further shows that the Kronecker product $K := K_1 \otimes K_2$ can be evaluated using a size $\text{poly}\, n$ depth $d_{\text{std}}$ uniform threshold circuit.

By Lemma 3.1 and the first part of Lemma B.3, the matrix product and division $QR_{j_1,j_2+(j_3-1)d}K^{\top}/d$ is evaluatable by $\text{poly}(n)$ size uniform threshold circuit with $3d_{\text{std}} + 2d_{\oplus}$ depth.

The exponential function $\exp()$ can be evaluated using Lemma B.4 by a size $\text{poly}\, n$ depth $d_{\exp}$ uniform threshold circuit.

Thus, the total required depth to compute the matrix $A$ is:

$$7d_{\text{std}} + 4d_{\oplus} + d_{\triangle} + d_{\exp}.$$

Any entry of $A_{i,j}, \forall i, j \in [n]$ can be evaluated in parallel, so the overall circuit size is $\text{poly}(n)$, and the total depth is $7d_{\text{std}} + 4d_{\oplus} + d_{\triangle} + d_{\exp}$.

The proof is thus concluded. $\square$

**Lemma E.8** (Single RoPE-based Attention Layer within $\mathsf{TC}^0$, informal version of Lemma 4.3). *For $p \leq \text{poly}(n)$, the Attn defined in Definition 2.18, can is evaluatable by a size $\text{poly}(n)$ depth $11d_{\text{std}} + 8d_{\oplus} + d_{\triangle} + d_{\exp}$ uniform threshold circuit.*

*Proof.* To evaluate Attn, the multiplication of the matrices $D^{-1}$, $A$, and $V$ is required. Initially, $D := \text{diag}(A\mathbf{1}n)$ can be evaluated by $\text{poly}(n)$ size uniform threshold circuit with $d_{\oplus}$ depth, as established in Part 3 of Lemma B.2. The matrix $A$ requires a circuit with $7d_{\text{std}} + 4d_{\oplus} + d_{\triangle} + d_{\exp}$ depth, according to Lemma E.7.

Next, the evaluation of $V := V_1 \oslash V_2$ is carried out in depth $d_{\text{std}}$, as per Lemma E.2. The multiplication of $A$ and $V$ is performed by $\text{poly}(n)$ size uniform threshold circuit of $d_{\text{std}} + d_{\oplus}$ depth, based on Lemma 3.1.

Lastly, the multiplication $D^{-1} \cdot AV$ is evaluated by performing division in parallel, which is implemented by $\text{poly}(n)$ size uniform threshold circuit of $d_{\text{std}} + d_{\oplus}$ depth, as per Part 1 of Lemma B.2.

Summing the circuit depths gives:

$$11d_{\text{std}} + 8d_{\oplus} + d_{\triangle} + d_{\exp}.$$

Because parallel operations can be conducted for each element, the attention operation $\text{Attn}(X)$ can be evaluated by a uniform threshold circuit with the required depth and size.

This concludes the proof. $\square$

**Lemma E.9** (Formal version of Lemma 4.4). *Consider the assumption that for every $i \in [m]$, $g_i$ in* TF *can be evaluated using* $\mathrm{poly}(n)$ *size uniform threshold circuit with a constant depth $d_g$. When* $p \leq \mathrm{poly}(n)$*, the* RoPE-*based tensor attention* TF*, as specified in Definition 2.21, can be evaluated by* $\mathrm{poly}(n)$ *size uniform threshold circuit of depth* $(m+1)d_g + 11md_{\mathrm{std}} + 8md_\oplus + m(d_\triangle + d_{\exp})$.

*Proof.* Under the given assumption, for every $i \in [m]$, $g_i$ can be evaluated by $\mathrm{poly}(n)$ size uniform threshold circuit having constant $d_g$ depth.

Moreover, from Lemma E.8, it follows that each $\mathsf{Attn}_i$ is evaluatable by $\mathrm{poly}(n)$ size uniform threshold circuit with depth $8d_{\mathrm{std}} + 6d_\oplus + d_\triangle + d_{\exp} + 1$.

To approximate $\mathsf{TF}(X)$, it is required to evaluate $g_0, g_1, \ldots, g_m$ and $\mathsf{Attn}_1, \ldots, \mathsf{Attn}_m$. As a result, the total depth of the $\mathrm{poly}(n)$ size circuit is $(m+1)d_g + 11md_{\mathrm{std}} + 8md_\oplus + m(d_\triangle + d_{\exp})$.

This concludes the proof. $\square$

**Theorem E.10** (Formal version of Theorem 4.5). *Assume that $\forall i \in [m]$, $g_i$ in* TF *can be computed using* $\mathrm{poly}(n)$ *size uniform threshold circuit of constant depth $d_g$. The* RoPE-*based tensor attention* TF*, as defined in Definition 2.21, is simulatable by uniform* $\mathsf{TC}^0$ *circuit family when $d \leq O(n), p \leq \mathrm{poly}(n)$, and $m \leq O(1)$.*

*Proof.* According to Lemma E.9, we have $m = O(1)$, the $O(\mathrm{poly}(n))$ bounded circuit used to compute $\mathsf{TF}(X)$ has a depth given by

$$(m+1)d_g + 11md_{\mathrm{std}} + 8md_\oplus + m(d_\triangle + d_{\exp}),$$

which bounded by $O(\mathrm{poly}(n))$. Thus, based on the definition of $\mathsf{TC}^0$, it follows that the uniform $\mathsf{TC}^0$ circuit family can approximate RoPE-based tensor attention Transformer.

The proof is complete. $\square$

# F    MISSING PROOFS IN SECTION 5

**Theorem F.1** (Formal version of Theorem 5.9). *If* $\mathsf{TC}^0 \neq \mathsf{NC}^1$, $O(1)$ *layers* RoPE-*based tensor attention Transformer with $d \leq O(n)$ hidden dimension,* $\mathrm{poly}(n)$ *precision is incapable of solving the fixed membership problem.*

*Proof.* The proof follows from the combination of Theorem E.10, which provides a circuit complexity bound for RoPE-based tensor attention Transformers, and Proposition 5.5, which establishes that the fixed membership problem for recognizing morphisms over finite words is $\mathsf{NC}^1$-complete. Additionally, Fact 2.9, which outlines the hierarchy of circuit families, is also applied here. This completes the proof. $\square$

**Theorem F.2** (Formal version of Theorem 5.10). *Assuming* $\mathsf{TC}^0 \neq \mathsf{NC}^1$, *a* $O(1)$ *layers tensor attention Transformer with $d \leq O(n)$ hidden dimension, and* $\mathrm{poly}(n)$ *precision is not capable of solving the $(A_{F,r})^*$ closure problem.*

*Proof.* This follows directly from Theorem E.10, which establishes the circuit complexity bound for RoPE-based tensor attention Transformers, and Theorem 5.8, which asserts that the $(A_{F,r})^*$ closure problem is $\mathsf{NC}^1$-complete. Additionally, Fact 2.9 concerning the hierarchy of circuit families is also utilized. Thus, the proof is complete. $\square$

## LLM USAGE DISCLOSURE

LLMs were used only to polish language, such as grammar and wording. These models did not contribute to idea creation or writing, and the authors take full responsibility for this paper's content.

