# OpenReview forum: "Theoretical Constraints on the Expressive Power of RoPE-based Tensor Attention Transformers"
_ICLR.cc/2026/Conference — Submitted to ICLR 2026_

### Official Review · Reviewer_Bzdt · 2025-10-21

**Soundness:** 2
**Presentation:** 1
**Contribution:** 3
**Rating:** 2
**Confidence:** 3

**Summary:**

This paper proposes a formal analysis of the expressive power of Tensor Attention Transformers (TAT) using Rotary positional embeddings (RoPE). The results presented in this paper can be seen as extending the results in (Chiang, 2024) who showed that transformers belong to DLOGTIME-uniform TC0 in two directions: TAT and RoPE.

The main results showed in this paper are that (i) TAT w/ and w/o RoPE (poly(n) size, constant depth) are also in DLOGTIME-uniform TC0 (Thm 3.7 and 4.5) and (ii) TAT w/ and w/o RoPE (constant depth) cannot solve the fixed membership problem (Thm 5.6), nor the (A_F,r)* closure problem (Thm 5.7).

There is no experimental section, the contribution is focused on the theory.

**Strengths:**

- Extending the collection of results characterizing the expressiveness of architectures commonly used in ML using circuit complexity is a very fundamental and relevant contribution.

**Weaknesses:**

- The main weakness of the paper in my opinion is its lack of clarity. This lack of clarity is due to several factor:

  - structure of the paper: most of the space is spent on definitions and results taken from other papers. As consequence, not enough time is spent on explaining how the results presented here are novel and non-trivial extension of previous work, as well as why they are relevant. I understand that it is important to introduce the previous results on which your work built upon, and that some definitions are unavoidable to present complex architecture designs such as tensor attention. Still:
      - some definitions are standard and should be omitted or deferred to appendix (e.g. Def 2.21 defines a rotation matrix, Def 2.20 defines the softmax function)
      - some facts should be deferred to the appendix since they are not use in any meaningful way in the main paper as far as I see (e.g. Fact 2.19 is mentioned in Def 2.25 but not needed to state this definition)
      - I would suggest only keeping the results that are fundamental to understand your contribution in the main paper and moved all the others to the appendix (e..g. I don't think the 4 results in section 2.1 are needed in the main paper). This would allow you to spend more time discussing your contribution.

 - May of the definitions and lemmas are taken from previous work and have been slightly rephrased / edited in a way that, in my opinion, make them less clear (e.g. Def 2.1). I would suggest using the exact same formulation as in the original papers for definition that are directly borrowed, unless there is a clear reason not to do so.

 - The paper lacks rigour in some places, some notations are not introduced. E.g. Q_{j_1,*} in Def 2.4, A+ and [P] in Def 5.1, the less or equal sign used in the sup of Def 5.3, ||w|| in Def 5.4.

 - Section 5, presenting the hardness result, is particularly difficult to follow (at least for me). Many notations have not been properly introduced. From the definitions I was not able to understand the definition of the two problems. I wrote my questions related to Def 5.1 and 5.3 below. Regarding Def 5.3. I do not understand why this very abstract definition of the Kleene star is used. My intuition is that the standard definition would be sufficient here (L^* = \cup_{n\geq 0} L^n).

**Questions:**

- What were the key technical challenges in proving Thm 4.5? Are the proof techniques a direct adaptation of the ones used in (Chiang, 2024) or were there any specific aspects of RoPE or TAT that made the analysis more challenging?

- Def 5.1: what does uv^\omega \in [P] means? What is omega here? What does the notation [P] refer to?

- Def 5.3: Why do you use the def from Kuznetsov for the Kleene star? I find it quite confusing and I believe un-necessary to define the closure problem where the standard definition would suffice.

**Details Of Ethics Concerns:**

-

---

> ### Author Response · Authors · 2025-11-27
>
> We would like to sincerely thank the reviewer for acknowledging our fundamental contribution to Transformer expressiveness. Regarding your detailed and in-depth comments, we have **made substantial revisions to the PDF file** and addressed all points. Below is a summary of the changes:
>
> ### Weakness 1: Structure of the paper
> Thanks for pointing this out. Following your suggestions, we have moved several technical components to the appendix:
> - Definitions of the softmax function and rotation matrix (originally Definitions 2.20 and 2.21), now moved to Definitions C.1 and C.2.
> - The fact on the swap rule for tensor and matrix products (originally Fact 2.19), now Fact D.1.
> - The four $\mathsf{TC}^0$ results for floating-point computation (original Lemmas 2.3–2.6), now Lemmas B.2-B.5.
>
> ### Weakness 2: Lexical Difference between Our Definition and the Original Definition
> We have revised the definition of floating-point numbers to make it more consistent with Chiang’s formalism. Please see the updated Definition 2.1.
>
> ### Weakness 3 and Question 2: Notation Issues
> We would like to clarify that in Definition 2.17, the notation $Q_{j_1,*}$ refers to the $j_1$-th row of $Q$, which is standard. The star symbol here has no relation to other concepts like duality or optimality.
>
> For the remaining notation issues, we have thoroughly improved all previously undefined or unclear notations, particularly for the fixed membership problem. Please see Definitions 5.1-5.3 and 5.7 for the revised statements. To aid understanding, we have also added a concrete example for the fixed membership problem in Example 5.4.
>
> ### Weakness 4 and Question 3: Kleene Star Definition
> Thank you for the helpful suggestion. We agree that the simpler form of the Kleene Star definition improves clarity for a machine learning audience. We have updated the definition accordingly. Please check out Definition 5.6 for our updates.
>
> ### Question 1: Technical Novelty
> We have added a new discussion paragraph on page 8 comparing our work with [Chiang, 2024]. We kindly invite the reviewer to examine this update.
>
> Once again, we sincerely thank you for your constructive feedback. Please check the updated PDF, and feel free to let us know if you have any further questions.

---

### Official Review · Reviewer_NWQ8 · 2025-10-31

**Soundness:** 2
**Presentation:** 2
**Contribution:** 2
**Rating:** 4
**Confidence:** 2

**Summary:**

The authors discuss the circuit complexity of tensor attention along with its RoPE-equipped variants. They show that under polynomial precision, constant-depth, and big O linear hidden dimension, these attention modules still cannot solve the fixed membership problem or the (A_{F,r})^* closure problem. These results highlight the computational limitation of tensor-attention.

**Strengths:**

The theoretical results on the limitation of tensor-attention seem solid, though I’m not an expert in the subfield.

**Weaknesses:**

It might be better to also run some experiments on the fixed membership problem or the (A_{F,r})^* closure problem using tensor attention with RoPE. Otherwise it’s hard to tell if this is actually a limitation in practice. It’s hard to tell if the theoretical complexity bound is tight or loose and how does it actually manifest in the Transformer model empirically.

**Questions:**

N/A

---

> ### Author Response · Authors · 2025-11-27
>
> We would like to sincerely thank the reviewer for the positive feedback on our theoretical contribution. Below, we clarify several points to address your concerns.
>
> ### Weakness 1: Rationale of a Theory Paper
> Thank you for raising this point. We agree that additional experiments could make the paper more comprehensive. However, implementing our approach is beyond the scope of this work, and we plan to leave it for future research. Our paper aligns with a wide range of prior purely theoretical Transformer expressiveness papers published at top ML conferences [1,2,3,4,5], and we believe that a theory-focused contribution is appropriate and relevant for ICLR.
>
> ### Weakness 2: Tightness of Bounds
> Your question regarding the tightness of our bounds is insightful, and we appreciate the opportunity to elaborate. Our main result, Theorem 4.5 on page 8, shows that RoPE-based tensor attention Transformers fall within the $\mathsf{TC}^0$ complexity class. Consequently, several $\mathsf{NC}^1$-hard problems cannot be solved by this architecture, as further demonstrated in Theorems 5.6-5.7 on page 9.
> Let $\mathsf{TF}$ denote the RoPE-based tensor attention Transformer considered in this work. Our result can be expressed as $\mathsf{TF} \subseteq \mathsf{TC}^0$. To tighten this upper bound, one might consider $\mathsf{AC}^0$, a well-studied class strictly weaker than $\mathsf{TC}^0$ (see classical results such as Corollary 4.35 in [6]). However, we believe it is difficult for $\mathsf{TF}$ to lie in $\mathsf{AC}^0$, since even a simplified Transformer model, average hard attention transformers (AHAT), is not computable in $\mathsf{AC}^0$, as shown in Theorem 3 of [7]. Therefore, $\mathsf{TC}^0$ represents a reasonably tight and meaningful upper bound.
>
> ### Weakness 3: Practical Implications
>
> Recent theoretical studies [8,9] suggest that standard Transformers may struggle to capture word-to-word-to-word (i.e., word-triplet) interactions, since vanilla self-attention focuses on pairwise relations. Tensor attention [10,11] has been proposed as a promising direction to address this limitation.
>
> In this work, we study whether tensor attention with RoPE can fundamentally improve the reasoning ability of Transformers from a circuit-complexity perspective. Our result shows that tensor attention falls into the low-level class $\mathsf{TC}^0$. This suggests that, unlike other techniques such as looped Transformers [1,12], Chain-of-Thought (CoT) [2,13], or improved positional encodings (which have known benefits from a circuit perspective) [14], tensor attention may not offer the desired expressive improvement on its own. Our findings highlight that these existing techniques remain important even in architectures that incorporate tensor attention.
>
>
> ### References
>
> [1] Angeliki Giannou, Shashank Rajput, Jy-Yong Sohn, Kangwook Lee, Jason D. Lee, Dimitris Papailiopoulos. “Looped Transformers as Programmable Computers”. ICML 2023.
>
> [2] Juno Kim and Taiji Suzuki. “Transformers Provably Solve Parity Efficiently with Chain of Thought”. ICLR 2025.
>
> [3] Bo Chen, Xiaoyu Li, Yingyu Liang, Jiangxuan Long, Zhenmei Shi, Zhao Song, Jiahao Zhang. “Circuit complexity bounds for rope-based transformer architecture”. EMNLP 2025.
>
> [4] Jerry Yao-Chieh Hu, Wei-Po Wang, Ammar Gilani, Chenyang Li, Zhao Song, Han Liu. “Fundamental Limits of Prompt Tuning Transformers: Universality, Capacity and Efficiency”. ICLR 2025.
>
> [5] Jerry Yao-Chieh Hu, Maojiang Su, En-jui Kuo, Zhao Song, Han Liu. “Computational Limits of Low-Rank Adaptation (LoRA) Fine-Tuning for Transformer Models”. ICLR 2025.
>
> [6] Heribert Vollmer. “Introduction to circuit complexity: a uniform approach”. Springer Science & Business Media, 1999.
>
> [7] William Merrill, Ashish Sabharwal, Noah A. Smith. “Saturated Transformers are Constant-Depth Threshold Circuits”. TACL 2022.
>
> [8] Clayton Sanford, Daniel J. Hsu, Matus J. Telgarsky. “Representational Strengths and Limitations of Transformers”. NeurIPS 2023.
>
> [9] Jacob Pfau, William Merrill, Samuel R. Bowman. “Let’s Think Dot by Dot: Hidden Computation in Transformer Language Models”. COLM 2024.
>
> [10] Xindian Ma, Peng Zhang, Shuai Zhang, Nan Duan, Yuexian Hou, Dawei Song, Ming Zhou. “A Tensorized Transformer for Language Modeling”. NeurIPS 2019.
>
> [11] Josh Alman, Zhao Song. “How to capture higher-order correlations? generalizing matrix softmax attention to Kronecker computation”. ICLR 2024.
>
> [12] Liu Yang, Kangwook Lee, Robert D Nowak, Dimitris Papailiopoulos. “Looped Transformers are Better at Learning Learning Algorithms”. ICLR 2024.
>
> [13] Zhiyuan Li, Hong Liu, Denny Zhou, Tengyu Ma. “Chain of Thought Empowers Transformers to Solve Inherently Serial Problems”. ICLR 2024.
>
> [14] Songlin Yang, Yikang Shen, Kaiyue Wen, Shawn Tan, Mayank Mishra, Liliang Ren, Rameswar Panda, Yoon Kim. “PaTH Attention: Position Encoding via Accumulating Householder Transformations”. NeurIPS 2025.

---

> > ### Comment · Reviewer_NWQ8 · 2025-11-27
> >
> > Thank you for addressing my concerns! I understand that this is a theoretical paper and empirical results are outside of the scope of this paper.
> >
> > I want to update the rating to 5 but can't find a button to do so. I just want to leave this comment here in case this openreview software bug is not addressed in time.

---

### Official Review · Reviewer_ZK2f · 2025-11-03

**Soundness:** 3
**Presentation:** 2
**Contribution:** 2
**Rating:** 4
**Confidence:** 3

**Summary:**

This paper is an investigation into the computational complexity of Tensor Attention Transformers and their RoPE-based variants using circuit complexity theory. The work builds through individual components to complete multi-layer architectures. The authors analyze circuit depth and size requirements for computing the tensor-attention related operations, and show that this can be simulated by uniform TC0 circuirts with constant depth, size, and precision. The results are extended to RoPE-based tensor attention, and limitations are shown in terms of solving particular classes of problems.

**Strengths:**

- Rigorous treatment with detailed proofs, building circuit complexity analysis systematically from basic operations
- Clear statement of assumptions and conclusions
- Useful extension of circuit complexity analysis to tensor attention and RoPE which are modern architectures
- Complexity bounds for all components

**Weaknesses:**

- no empirical evaluation at all; no practical validation of key assumptions/conclusions. No ablations
- no guidance for practitioners. at times, unclear how to use insights for architecture design
- more discussion in terms of bound tightness and practical implications
- limited intuition and clarity of key takeaways while the paper is technically very dense
- The approach extends related work and adapts to tensor attention (somewhat incremental novelty in that respect)
- results apply only under constraints that may not hold in practice (constant layers etc.).
- Additionally (minor) make sure that notation is consistent/clear (e.g. 2.24 vs 2.25 definition)

**Questions:**

please see above limitations and weaknesses: e.g., how do constraints/assumptions link with practical transformer implementations, empirical validation, missing link to practitioners, and so on.

---

> ### Author Response · Authors · 2025-11-27
> **First response to reviewer Zk2f, Parts 1/2**
>
> We would like to sincerely thank the reviewer for acknowledging our theoretical rigor and detailed analysis of each model component. Here are our detailed responses to your concerns:
>
> ### Weakness 1: Rationale of Theory Paper
> Thank you for raising this point. We agree that additional experiments could make the paper more comprehensive. However, implementing our approach is beyond the scope of this work, and we plan to leave it for future research. Our paper aligns with a wide range of prior purely theoretical Transformer expressiveness papers published at top ML conferences [1,2,3,4,5], and we believe that a theory-focused contribution is appropriate and relevant for ICLR.
> ### Weakness 2 & Weakness 4: Practical Implications
>
> Recent theoretical studies [6,7] suggest that standard Transformers may struggle to capture word-to-word-to-word (i.e., word-triplet) interactions, since vanilla self-attention focuses on pairwise relations. Tensor attention [8,9] has been proposed as a promising direction to address this limitation.
>
> In this work, we study whether tensor attention with RoPE can fundamentally improve the reasoning ability of Transformers from a circuit-complexity perspective. Our result shows that tensor attention falls into the low-level class $\mathsf{TC}^0$. This suggests that, unlike other techniques such as looped Transformers [1,10], Chain-of-Thought (CoT) [2,11], or improved positional encodings (which have known benefits from a circuit perspective) [14], tensor attention may not offer the desired expressive improvement on its own. Our findings highlight that these existing techniques remain important even in architectures that incorporate tensor attention.
>
> ### Weakness 3: Tightness of Bounds
>
> Your question regarding the tightness of our bounds is insightful, and we appreciate the opportunity to elaborate. Our main result, Theorem 4.5 on page 8, shows that RoPE-based tensor attention Transformers fall within the $\mathsf{TC}^0$ complexity class. Consequently, several $\mathsf{NC}^1$-hard problems cannot be solved by this architecture, as further demonstrated in Theorems 5.6-5.7 on page 9.
> Let $\mathsf{TF}$ denote the RoPE-based tensor attention Transformer considered in this work. Our result can be expressed as $\mathsf{TF} \subseteq \mathsf{TC}^0$. To tighten this upper bound, one might consider $\mathsf{AC}^0$, a well-studied class strictly weaker than $\mathsf{TC}^0$ (see classical results such as Corollary 4.35 in [12]). However, we believe it is difficult for $\mathsf{TF}$ to lie in $\mathsf{AC}^0$, since even a simplified Transformer model, average hard attention transformers (AHAT), is not computable in $\mathsf{AC}^0$, as shown in Theorem 3 of [13]. Therefore, $\mathsf{TC}^0$ represents a reasonably tight and meaningful upper bound.
> ### Weakness 5: Importance of Tensor Attention
> Thanks for pointing this out. Tensor attention is a highly non-trivial and emerging type of Transformer architecture that addresses practical problems such as capturing word-to-word-to-word (higher-order) relations [6,7,8,9]. We believe analyzing this new architecture is highly impactful, and that this is an important problem.
>
> ### Weakness 6: Constant Layer Assumption
>
> Our main result (Theorem 3.7) assumes $O(1)$ model depth. This assumption is highly practical, as real‑world LLMs typically have a fixed number of layers (e.g., 24 or 32) and cannot grow arbitrarily deep after training. To boost expressivity via increasing model depth, one could use CoT with $O(\log^k n)$ reasoning steps [11] or $O(\log^k n)$ looped Transformers [1,10]. By definition of $\mathsf{TC}^k$, a trivial corollary is that Transformers with $O(\log^k n)$ depth become $\mathsf{TC}^k$. However, these settings differ from the tensor attention Transformers studied in this work, and may not affect the practicality of our constant layer setting.
>
> ### Weakness 7: Notation Problems
> Thank you for carefully checking the mathematical notation. This is greatly appreciated. We have updated the PDF for Definition 2.25 (now Definition 2.18) to ensure it is clearer and more consistent with Definition 2.24 (now Definition 2.17).

---

> ### Author Response · Authors · 2025-11-27
> **First response to reviewer Zk2f, Parts 2/2**
>
> ### References
>
> [1] Angeliki Giannou, Shashank Rajput, Jy-Yong Sohn, Kangwook Lee, Jason D. Lee, Dimitris Papailiopoulos. “Looped Transformers as Programmable Computers”. ICML 2023.
>
> [2] Juno Kim and Taiji Suzuki. “Transformers Provably Solve Parity Efficiently with Chain of Thought”. ICLR 2025.
>
> [3] Bo Chen, Xiaoyu Li, Yingyu Liang, Jiangxuan Long, Zhenmei Shi, Zhao Song, Jiahao Zhang. “Circuit complexity bounds for rope-based transformer architecture”. EMNLP 2025.
>
> [4] Jerry Yao-Chieh Hu, Wei-Po Wang, Ammar Gilani, Chenyang Li, Zhao Song, Han Liu. “Fundamental Limits of Prompt Tuning Transformers: Universality, Capacity and Efficiency”. ICLR 2025.
>
> [5] Jerry Yao-Chieh Hu, Maojiang Su, En-jui Kuo, Zhao Song, Han Liu. “Computational Limits of Low-Rank Adaptation (LoRA) Fine-Tuning for Transformer Models”. ICLR 2025.
>
> [6] Clayton Sanford, Daniel J. Hsu, Matus J. Telgarsky. “Representational Strengths and Limitations of Transformers”. NeurIPS 2023.
>
> [7] Jacob Pfau, William Merrill, Samuel R. Bowman. “Let’s Think Dot by Dot: Hidden Computation in Transformer Language Models”. COLM 2024.
>
> [8] Xindian Ma, Peng Zhang, Shuai Zhang, Nan Duan, Yuexian Hou, Dawei Song, Ming Zhou. “A Tensorized Transformer for Language Modeling”. NeurIPS 2019.
>
> [9] Josh Alman, Zhao Song. “How to capture higher-order correlations? generalizing matrix softmax attention to Kronecker computation”. ICLR 2024.
>
> [10] Liu Yang, Kangwook Lee, Robert D Nowak, Dimitris Papailiopoulos. “Looped Transformers are Better at Learning Learning Algorithms”. ICLR 2024.
>
> [11] Zhiyuan Li, Hong Liu, Denny Zhou, Tengyu Ma. “Chain of Thought Empowers Transformers to Solve Inherently Serial Problems”. ICLR 2024.
>
> [12] Heribert Vollmer. “Introduction to circuit complexity: a uniform approach”. Springer Science & Business Media, 1999.
>
> [13] William Merrill, Ashish Sabharwal, Noah A. Smith. “Saturated Transformers are Constant-Depth Threshold Circuits”. TACL 2022.

---

### Official Review · Reviewer_S3FE · 2025-11-07

**Soundness:** 3
**Presentation:** 2
**Contribution:** 3
**Rating:** 6
**Confidence:** 1

**Summary:**

This paper analyzes the circuit complexity of the Tensor Attention Transformer and extends it to its RoPE-based Tensor Attention variants. With the analysis, their work provides theoretical understanding of self-attention architectures and computational boundaries. These findings highlight a gap between the empirical performance and theoretical constraints of Tensor Attention and RoPE-based Tensor Attention Transformers, offering insights that could guide the development of more theoretically grounded approaches to Transformer
model design and scaling.

**Strengths:**

1. The results of this paper can provide valuable insights into existing structures.

2. Using circuit complexity theory, this paper analyzes tensor attention and rotary position embedding (RoPE).'

3. The self-attention structure is crucial, and analyzing its expressiveness will exert a profound and long-lasting impact.

4. In my opinion, this work is novel for the related area.

**Weaknesses:**

1. It's hard to understand some concepts, such as CIRCUIT COMPLEXITY.

**Questions:**

How can this theroy result motivate further improvements to the self-attention structure?

---

> ### Author Response · Authors · 2025-11-27
>
> Thanks for the positive and encouraging review. Below, we provide our detailed responses.
>
> ### Weakness 1: Background in circuit complexity
>
> We appreciate the request for more background. Here, we briefly summarize the key circuit-complexity concepts used in our paper. Please refer to textbooks like [1,2] for more details.
>
> **Circuit.**
> In theoretical computer science, a circuit is a collection of interconnected gates that process input bits and produce an output. Each gate performs a simple logical operation (e.g., AND, OR, NOT). Circuits can have different fan-in (number of inputs per gate) and fan-out (number of outputs). A nonuniform circuit family allows a different circuit for each input size $n$, while a uniform family requires that the circuits be constructible by an efficient algorithm (e.g., DLOGTIME-uniform or L-uniform), as defined in our paper.
>
> **Gate.**
> Standard Boolean gates such as AND, OR, and NOT usually have constant fan-in (typically 2). Unbounded fan-in circuits allow AND and OR gates to take an arbitrary number of inputs. MAJORITY is a stronger gate that outputs 1 when more than half of its inputs are 1. Threshold gates like MAJORITY can compute certain functions more efficiently than simple Boolean gates, making them important in constant-depth circuit classes.
>
> **Circuit complexity.**
> Circuit complexity studies how large (size) and how deep (depth) a circuit must be to compute a function. A problem is considered easy if it can be computed by circuits that are small and shallow, and hard if it requires large or deep circuits. Prior work has used circuit size, depth, and gate types to define complexity classes and compare the expressive power of different models.
>
> **Difference between $\mathsf{TC}^0$ and $\mathsf{NC}^1$.**
> Both $\mathsf{TC}^0$ and $\mathsf{NC}^1$ use polynomial-size circuits, but they differ in depth and gate types.
> - $\mathsf{NC}^1$ allows logarithmic depth and uses only bounded fan-in Boolean gates.
> - $\mathsf{TC}^0$ requires constant depth but permits more powerful unbounded fan-in threshold gates (such as MAJORITY).
> Although $\mathsf{TC}^0$ has stronger gates, its constant-depth restriction makes it strictly less expressive than $\mathsf{NC}^1$. As a result, $\mathsf{TC}^0 \subseteq \mathsf{NC}^1$. This difference helps us characterize the theoretical limits of attention variants.
>
> ### Question 1: Implications of the main results
>
> Recent theoretical studies [3,4] suggest that standard Transformers may struggle to capture word-to-word-to-word (i.e., word-triplet) interactions, since vanilla self-attention focuses on pairwise relations. Tensor attention [3,5,6] has been proposed as a promising direction to address this limitation.
>
> In this work, we study whether tensor attention with RoPE can fundamentally improve the reasoning ability of Transformers from a circuit-complexity perspective. Our result shows that tensor attention falls into the low-level class $\mathsf{TC}^0$. This suggests that, unlike other techniques such as looped Transformers [7,8], Chain-of-Thought (CoT) [9, 10], or improved positional encodings (which have known benefits from a circuit perspective) [11], tensor attention may not offer the desired expressive improvement on its own. Our findings highlight that these existing techniques remain important even in architectures that incorporate tensor attention.
>
> ### References
>
> [1] Heribert Vollmer. “Introduction to circuit complexity: a uniform approach”. Springer Science & Business Media, 1999.
>
> [2] Sanjeev Arora, Boaz Barak. “Computational complexity: a modern approach”. Cambridge University Press, 2009.
>
> [3] Clayton Sanford, Daniel J. Hsu, Matus J. Telgarsky. “Representational Strengths and Limitations of Transformers”. NeurIPS 2023.
>
> [4] Jacob Pfau, William Merrill, Samuel R. Bowman. “Let’s Think Dot by Dot: Hidden Computation in Transformer Language Models”. COLM 2024.
>
> [5] Xindian Ma, Peng Zhang, Shuai Zhang, Nan Duan, Yuexian Hou, Dawei Song, Ming Zhou. “A Tensorized Transformer for Language Modeling”. NeurIPS 2019.
>
> [6] Josh Alman, Zhao Song. “How to capture higher-order correlations? generalizing matrix softmax attention to Kronecker computation”. ICLR 2024.
>
> [7] Angeliki Giannou, Shashank Rajput, Jy-Yong Sohn, Kangwook Lee, Jason D. Lee, Dimitris Papailiopoulos. “Looped Transformers as Programmable Computers”. ICML 2023.
>
> [8] Liu Yang, Kangwook Lee, Robert D Nowak, Dimitris Papailiopoulos. “Looped Transformers are Better at Learning Learning Algorithms”. ICLR 2024.
>
> [9] Zhiyuan Li, Hong Liu, Denny Zhou, Tengyu Ma. “Chain of Thought Empowers Transformers to Solve Inherently Serial Problems”. ICLR 2024.
>
> [10] Juno Kim and Taiji Suzuki. “Transformers Provably Solve Parity Efficiently with Chain of Thought”. ICLR 2025.
>
> [11] Songlin Yang, Yikang Shen, Kaiyue Wen, Shawn Tan, Mayank Mishra, Liliang Ren, Rameswar Panda, Yoon Kim. “PaTH Attention: Position Encoding via Accumulating Householder Transformations”. NeurIPS 2025.

---

### Author Response · Authors · 2025-12-03
**Summary of the discussion period**

Dear Area Chair,

Given the unprecedented situation regarding the OpenReview information leak and ICLR’s decision to revert pre-rebuttal scores and reassign ACs, we would like to summarize our interactions with the reviewers during the discussion period to ensure that our responses are clearly documented.

- **Reviewer S3FE**: The reviewer requested (1) additional background on circuit complexity and (2) clarification of the practical implications of our theoretical results. We addressed these points extensively in the rebuttal. The reviewer **maintains a positive score of 6**.

- **Reviewer Zk2f**: The reviewer raised questions about (1) the rationale for a theory paper at ICLR, (2) practical implications, (3) tightness of bounds, (4) importance of tensor attention, (5) assumptions, and (6) notation issues. We **revised the PDF** to address these concerns, including updating Definition 2.25 (now Definition 2.18) for consistency with Definition 2.24 (now Definition 2.17). The reviewer **did not respond during the discussion period**.

- **Reviewer NWQ8**: The reviewer asked about (1) the rationale for a theory paper at ICLR, (2) practical implications, and (3) tightness of bounds. We responded carefully to each point. The reviewer **increased the score to a positive score of 6**.

- **Reviewer Bzdt**: The reviewer provided insightful suggestions on clarity of the statements. We **significantly improved the PDF** in response, especially on pages 3-6, 8-9, and 14-15. The reviewer **did not respond during the discussion period**.

We believe we have thoroughly addressed all reviewer comments and made substantial improvements to the paper based on the discussion. We hope this summary clarifies the reviewers’ positions following the rebuttal and highlights our efforts to incorporate their feedback. Thank you very much for your consideration.

---

### Meta-Review · Area_Chair_LT1D · 2026-01-13

**Summary:**

The reviewers broadly agree that the paper presents a technically careful theoretical analysis of tensor attention and RoPE-based tensor attention transformers using circuit complexity, extending prior expressiveness results for standard Transformers. Several reviewers acknowledge the soundness and rigor of the proofs (e.g., Reviewer S3FE: “the work provides theoretical understanding of self-attention architectures and computational boundaries”; Reviewer Zk2f: “rigorous treatment with detailed proofs”).

However, the dominant concerns across reviews relate to (i) lack of practical relevance and validation, (ii) unclear implications for real transformer design, and (iii) presentation and clarity issues that significantly hinder accessibility, especially in the hardness results. Multiple reviewers explicitly question whether the strong theoretical assumptions (e.g., constant depth, polynomial precision, linear/sublinear hidden dimension) meaningfully reflect practical transformer settings (Reviewer Zk2f, Reviewer NWQ8), and whether the resulting TC0 upper bound manifests as a real limitation in practice.

While the authors provide extensive rebuttal explanations and revise the manuscript to improve clarity, the rebuttal does not introduce new evidence—theoretical or empirical—that resolves the central concerns about practical significance, tightness of the bounds in realistic regimes, or actionable guidance for practitioners. As a result, despite acknowledged technical merit, the paper falls short of the bar for acceptance at ICLR due to limited impact beyond a narrow theoretical setting and remaining clarity and relevance gaps.

**Reviewer Concerns:**

Concerns substantially addressed by the rebuttal

- Background and terminology clarity (Reviewer S3FE):
The authors added a clear explanation of circuit complexity concepts (e.g., TC0 vs. NC1) and provided textbook references. This directly addresses the reviewer’s comment that “it’s hard to understand some concepts, such as circuit complexity,” and improves accessibility for non-experts.

- Notation inconsistencies and definitional issues (Reviewer Zk2f, Reviewer Bzdt):
The authors revised inconsistent definitions (e.g., Definition 2.24 vs. 2.25), clarified previously undefined symbols, and simplified the Kleene star definition in response to Reviewer Bzdt’s detailed critique. The commitment to moving standard definitions and auxiliary lemmas to the appendix is appropriate and likely improves readability.

- Justification of a theory-only contribution (Reviewer Zk2f, Reviewer NWQ8):
The rebuttal reasonably explains that the paper is intended as a purely theoretical contribution, consistent with prior expressiveness work at ICLR/NeurIPS. Reviewer NWQ8 explicitly acknowledged this explanation and indicated acceptance of the lack of experiments.

Concerns that remain outstanding

- Lack of empirical or practical validation (Reviewer Zk2f, Reviewer NWQ8):
Multiple reviewers requested at least minimal experimental evidence or concrete demonstrations showing whether the theoretical TC0 limitation meaningfully constrains tensor attention models in practice. The rebuttal does not provide any new validation, instead deferring experiments to future work. As Reviewer Zk2f noted, without such evidence it remains “unclear how to use the insights for architecture design,” and whether the bounds are tight or loose in realistic regimes.

- Unclear practical implications for model design (Reviewer S3FE, Reviewer Zk2f):
Although the authors argue that tensor attention alone does not improve expressivity beyond TC0 and contrast it with techniques like looped transformers or Chain-of-Thought, this remains a high-level conceptual claim. Reviewers explicitly asked how the results “motivate further improvements to the self-attention structure” (Reviewer S3FE), but the rebuttal does not translate the theory into concrete, actionable design guidance.

- Relevance of assumptions to real transformers (Reviewer Zk2f):
The rebuttal argues that constant depth is “highly practical” because real LLMs have fixed numbers of layers. However, this does not fully address the reviewer’s concern that circuit-complexity asymptotics may fail to capture expressivity gains from width, precision, training dynamics, or multi-step inference in practice. This conceptual gap remains unresolved.

- Incremental novelty relative to prior work (Reviewer Zk2f, Reviewer Bzdt):
While the authors added discussion comparing their results to prior work (e.g., Chiang, 2024), reviewers questioned whether the extension to tensor attention and RoPE introduces fundamentally new technical insights versus an adaptation of existing techniques. The rebuttal asserts novelty but does not fully dispel the perception of incremental contribution.

**Reviewer Scores:**

- Reviewer S3FE (initial rating: 6, confidence: 1)
This reviewer already maintained a positive score after the rebuttal and did not raise further objections. Given their low confidence and the rebuttal clarifications, the score would likely remain at 6, though still with limited weight due to uncertainty.

- Reviewer Zk2f (initial rating: 4)
Although the rebuttal addresses notation and scope questions, the reviewer’s core concerns—lack of empirical validation, limited practitioner guidance, and incremental novelty—remain largely unresolved. It is unlikely this reviewer would substantially revise their assessment.

- Reviewer NWQ8 (initial rating: 4)
After the rebuttal, this reviewer explicitly stated they intended to update the score to 5 but were unable to do so due to ICLR's rule. This change reflects acceptance of the theory-only scope, but still does not indicate strong endorsement.

- Reviewer Bzdt (initial rating: 2)
Despite the authors’ promised revisions, this reviewer raised deep concerns about clarity, rigor, and presentation of the core hardness results, as well as questions about technical novelty. Without confirmation that the revised manuscript fully resolves these issues, it is unlikely the reviewer would raise their score substantially; the score would likely remain low (around 2–4).

---

### Decision · Program_Chairs · 2026-01-26

Reject